# Determinants of voluntary disclosure: An empirical analysis of financial, market, and organizational factors

Yuichiro Nakai[ID][1]*, Mitsuo Yoshida[ID][2]

**1** Degree Programs in Business Sciences, University of Tsukuba, Tokyo, Japan, **2** Institute of Business Sciences, University of Tsukuba, Tokyo, Japan

* s2440411@u.tsukuba.ac.jp

## Abstract

This study systematically examines the determinants of voluntary disclosure by listed companies in Japan, where non-financial disclosure is not legally mandated. Using quarterly data from 5,915 companies between 2009 and 2024, a logistic regression analysis was conducted to assess the influence of financial indicators, company characteristics, stock exchange, industry classification, and shareholder composition. The results reveal that ISO certification (ISO 14001 and ISO 45001) significantly promotes voluntary disclosure. Firm size and listing on a prime market are positively associated with disclosure, whereas foreign market listings tend to suppress disclosure—contrary to conventional theories. Additionally, manufacturing and IT sectors are proactive in disclosure, whereas the agriculture, forestry, fishing, and mining sectors are more reluctant. Shareholder composition also plays a crucial role, with financial institution shareholders promoting disclosure and corporate shareholders suppressing it. By leveraging Japan's unique voluntary disclosure environment, this study offers new insights into corporate transparency and the strategic design of disclosure policies.

## 1. Introduction

"Why do companies choose to disclose information they are not legally obligated to disclose?" "What is behind the voluntary CSR and ESG reporting by many Japanese companies, despite the lack of legal obligation?" Due to the changing nature of motivation and scope of disclosure over time, there are many studies on disclosure, and it is essential to examine the current disclosure issues. Therefore, we examine this issue by organizing corporate disclosure into two types: statutory and voluntary shown in Table 1.

Statutory disclosures are disclosures of financial information required by law or regulation and include annual reports, financial statements, and corporate governance reports. These are intended to ensure transparency to investors and

**Data availability statement:** All relevant data are within the manuscript and its Supporting Information files.

**Funding:** The author(s) received no specific funding for this work.

**Competing interests:** The authors have declared that no competing interests exist.

regulators. In recent years, non-financial information has been increasingly recognized as an increasingly important part of statutory disclosure [1,2], but its scope remains limited. On the other hand, companies may choose to use voluntary disclosures to supplement statutory disclosures. Voluntary disclosures are additional information not required by Generally Accepted Accounting Principles (GAAP) or specific regulations [3] and include Corporate Social Responsibility (CSR) reports and Environmental, Social, and Governance (ESG) reports, and integrated reporting, as companies voluntarily provide financial and non-financial information not required by statutory disclosure. Voluntary disclosure is becoming increasingly important in today's capital markets because it helps companies better identify the social responsibilities [4] they assume and activities that create value for investors and reduce information asymmetries between shareholders and management [5]. While voluntary disclosure is important in building investor and stakeholder trust, it also entails a conflict between transparency and maintaining competitive advantage, due to the need to avoid providing too much information. In addition, it is difficult to directly measure a company's emotional and internal decision-making processes, and this poses a challenge when understanding corporate behavior from the outside. Therefore, externally available objective data must be used to infer corporate decision-making.

However, few studies have conducted exploratory and comprehensive analyses of the factors that encourage companies to actively disclose information and the factors that discourage them from doing so.

Japan is unique in this regard. Companies there are not legally required to disclose non-financial information, such as CSR and ESG reporting. Instead, they are free to make their own choices. This allows for interesting comparisons between companies that disclose such information and those that do not. This unique situation makes Japan an important case in international comparative studies. It is also very important to analyze how voluntary disclosure is reflected in corporate strategy. We can do this by comparing it with mandatory disclosure standards in other countries, such as South Africa and the EU. The purpose of this study is to explore and clarify the decision-making factors that influence Japanese Companies' choice of voluntary

**Table 1. Classification of corporate disclosure and the position of voluntary disclosure in this report.**

| Disclosure (major category) | Disclosure (Medium Category) | Classification. | Example | Contents | Nature |
|---|---|---|---|---|---|
| Statutory disclosure (broadly defined) | Statutory disclosure (narrowly defined) | Legal, etc. (hard law) | Annual Securities Report, Financial Statements | Focus on financial information | IR |
| | Statutory disclosure (Other) | Market rules, etc. (soft law) | Corporate Governance Report | Financial and non-financial information | |
| Voluntary disclosure (broadly defined) | Voluntary disclosure (supplemental to statutory disclosure) | Public Relations Activities for the Press | Press Release | Financial and non-financial information, Product information, etc. | IR-oriented |
| | Voluntary Disclosure (in this paper) | corporate activities | CSR Reporting, ESG Reporting, SDGs Reporting, Integrated Reporting, Sustainability Reporting | | |
| | Voluntary Disclosure (Other) | propaganda activities | Commercials, etc. | Product information, etc. | PR |

disclosure. Specifically, we systematically examine how these factors affect voluntary disclosure from five perspectives: financial indicators, Company characteristics, listed market, industry classification, and shareholder composition.

The period analyzed in this study is the quarterly data from March 2009 to March 2024, covering a total of 5,915 Japanese listed companies during this period. Based on these data, we used logistic regression analysis to examine the effects of financial indicators, Company characteristics, listing market, industry classification, and shareholder composition.

As a result, the following key findings were obtained.

1. Academic originality and novelty

- Foreign Market Listings Suppress Voluntary Disclosure: While conventional theory associates foreign market listings with increased transparency and disclosure, this study finds the opposite.

- Companies with higher Price-to-Earnings (P/E) ratios, indicating market expectations for growth, are less likely to voluntarily disclose due to competitive reasons is a new finding that contradicts conventional theory.

- ISO certifications, such as ISO 14001 (environmental management systems) and ISO 45001 (occupational health and safety management systems), actively promote voluntary disclosure.

2. Consistency with research objectives

- Company size (total assets and sales) promotes voluntary disclosure: The tendency for larger companies to actively engage in voluntary disclosure was clearly confirmed.

- Increased voluntary disclosure activity in prime markets: The results show that differences in market segmentation have a strong influence on Companies' voluntary disclosure behavior.

- Trends by industry category: The manufacturing and information and communications industries were highly proactive, while the agriculture, forestry, fisheries, and mining industries showed a markedly restrained trend.

3. Practical and policy contributions

- Demand for transparency in the prime market: The increased activity of voluntary disclosure in the prime market reflects the demand for transparency from investors and has policy implications.

- Differences by shareholder composition: Voluntary disclosure is promoted when financial institutions are the major shareholders, while it is suppressed when there are many other corporate shareholders.

- Practical impact of ISO certification: Environmental and occupational safety initiatives are specifically linked to the company's information disclosure strategy.

The novelty of this study lies in its comprehensive analysis of corporate decision making, taking advantage of Japan's unique voluntary disclosure environment. In particular, the finding that listing on a foreign market suppresses voluntary disclosure, contrary to existing theory, is a unique result of this study. These results provide new insights into the voluntary disclosure behavior of Japanese companies, deepen our understanding of corporate strategy and fulfillment of social responsibility in a non-regulatory environment, and provide practical implications for international disclosure standards and sustainability strategies.

## 2. Literature review

In this chapter, we organize disclosure theory and prior research to develop a theoretical framework for identifying factors that influence voluntary disclosure by Japanese Companies. The literature review has two aspects: prior knowledge and

prior research. Since the purpose of this paper is to provide a comprehensive, exploratory analysis of the factors affecting voluntary disclosure, the selected literature provides a basis for developing theories and hypotheses supporting voluntary disclosure from the following perspectives. First, we introduce some specific theories, country systems, and the Japanese situation that support our study. Second, we introduce previous studies relevant to our study and identify in detail the theoretical and empirical bases for each of our hypotheses.

## 2.1. Background knowledge

### 2.1.1. Disclosure theory and prior research.
Based on agent theory, companies are encouraged to disclose information to eliminate information asymmetries and increase transparency [6]. However, excessive disclosure also carries the risk of losing a competitive advantage, so careful judgment is required [7,8]. There is much discussion about how corporate disclosure affects strategy and its intentions. Some have noted that "disclosure is one of the most important decisions in business" and that it affects corporate strategy, stakeholders, and even market valuation [9]. In other words, it is important to note that information is not simply a matter of disclosure. There is also debate about the roles and interactions between voluntary and statutory disclosure. For example, while there is little coordination between investor relations (IR) and general public relations (PR) activities, it has been pointed out that an appropriate combination of voluntary and statutory disclosure may create synergy [10]. Some suggest that integrated reporting (IR), which deals with financial and non-financial information, combines traditional financial accounting with sustainability and corporate governance issues to enhance decision usefulness and play a key role in business management and stakeholder relations management [11]. These studies suggest that corporate disclosure is heavily influenced by strategy, business environment, and even market expectations. Initiative-taking disclosure increases transparency and strengthens trust with investors and regulators. However, excessive disclosure also carries the risk of losing a competitive advantage, so prudent and strategic decisions must be made.

### 2.1.2. Comparison of International Disclosure Systems.
Currently, due to differences in the legal and other systems of various countries, international differences in the range of information disclosures are recognized. For example, South Africa has required companies listed on the Johannesburg Stock Exchange to submit integrated reports since 2010 (King Code Chapter 9). In France, CSR reporting has been mandatory since 2001 [12], and the EU passed a law requiring the disclosure of non-financial information in 2014 [13]. In Italy, large companies with more than five hundred employees are required to disclose SDG-related information from 2017 [14], among other differences. In Europe, the Value Reporting Foundation (VRF), established by the merger of the International Integrated Reporting Council (IIRC) and the Sustainability Accounting Standards Board (SASB), and the International Financial Reporting Standards (IFRS) Foundation, which develops global accounting standards (https://www.ifrs.org/news-and-events/news/2022/08/ifrs -foundation-completes-consolidation-with-value-reporting-foundation/), and international standards for ESG information have been developed (https://www.ifrs.org/news-and-events/news/ 2022/08/ifrs-foundation-completes-consolidation-with-value-reporting-foundation/). This consolidation will improve the consistency and comparability of information and meet the requirements of international investors.

According to the author's tabulations, Fig 1 illustrates the increasing trend of Integrated Reports and Company Name Reports, alongside a noticeable decline in Annual Reports. This indicates that Japanese companies have historically prepared voluntary reports under various names such as "Annual Report," "CSR Report," and "ESG Report" to align with international requirements appropriately. Recently, companies have increasingly discontinued the use of "CSR Reports" and "ESG Reports," opting instead to consolidate information into "Integrated Reports" and "Company Reports."

In Japan, there is no legal obligation for companies to disclose environmental and social responsibility information, and companies are free to choose whether or not to do so on a voluntary basis. CSR and ESG reporting are the most common forms of voluntary disclosure by Japanese companies, and most of them are provided in the form of integrated reports and company reports. These reports are unique in that they conform to international standards such as the Global

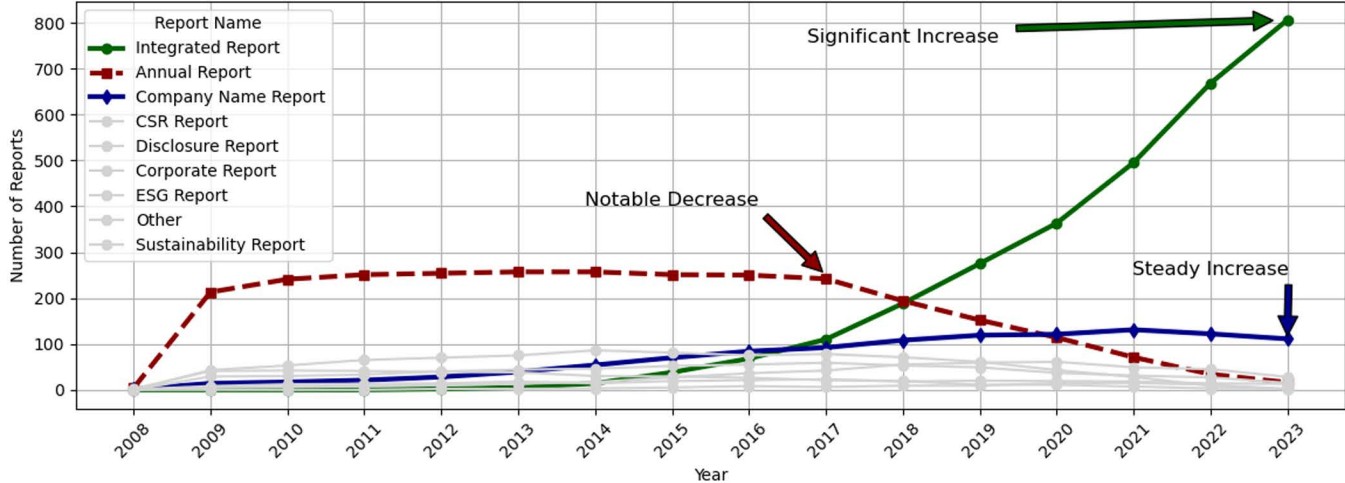

**Fig 1. Trends in voluntary disclosure report names for Japanese Listed Companies 2008-2023.** This figure highlights the significant increase in the number of Integrated Reports and Company Name Reports, as well as the notable decrease in the number of Annual Reports issued by Japanese listed companies from 2008 to 2023.

Reporting Initiative (GRI) and International Financial Reporting Standards (IFRS), while at the same time reflecting regional characteristics and cultural backgrounds. The voluntary nature of the standard is a feature of the environment. This environment of voluntary nature makes Japan an ideal case for exploring the factors and motivations that influence voluntary disclosure behavior.

Fig 2 illustrates trends in the number of listed companies and voluntary disclosers in Japan from 2009 to 2024. The data reveals a steady increase in companies engaging in voluntary disclosure, although not all listed companies actively participate. Consequently, this study seeks to identify the underlying factors influencing companies' decisions to undertake voluntary disclosure.

Voluntary disclosure is part of corporate strategy and a means of demonstrating corporate transparency and social responsibility to investors and stakeholders. However, it remains unclear which factors influence voluntary disclosure. While there has been research on the impact of the combination of statutory and voluntary disclosure on corporate valuation, there is limited knowledge on how corporate characteristics and market differences are reflected in voluntary disclosure. In addition, since decisions regarding voluntary disclosure are made internally within a company, it is difficult to study the process from the outside. Nevertheless, it is clear that Companies make voluntary disclosure decisions in consideration of the business environment. The purpose of this study is to analyze the characteristics of companies that disclose voluntarily and those that do not, based on publicly available information such as corporate websites and financial reports, and to identify the factors that contribute to voluntary disclosure.

## 2.2. Previous studies and research hypotheses

This study defines voluntary disclosure as additional information that is not required by Generally Accepted Accounting Principles (GAAP) or specific regulations but is voluntarily disclosed by companies. Specifically, CSR reporting, ESG reporting, SDGs reporting, integrated reporting, and sustainability reporting prepared by Japanese listed companies fall into this category, and while these reports are not legally required, they are positioned as important means of demonstrating corporate strategy and social responsibility. The purpose of this study is to elucidate the factors that lead Japanese listed companies to choose to disclose these voluntary disclosures. Specifically, we aim to identify the factors that influence voluntary corporate disclosure behavior from the five perspectives of financial indicators, corporate characteristics,

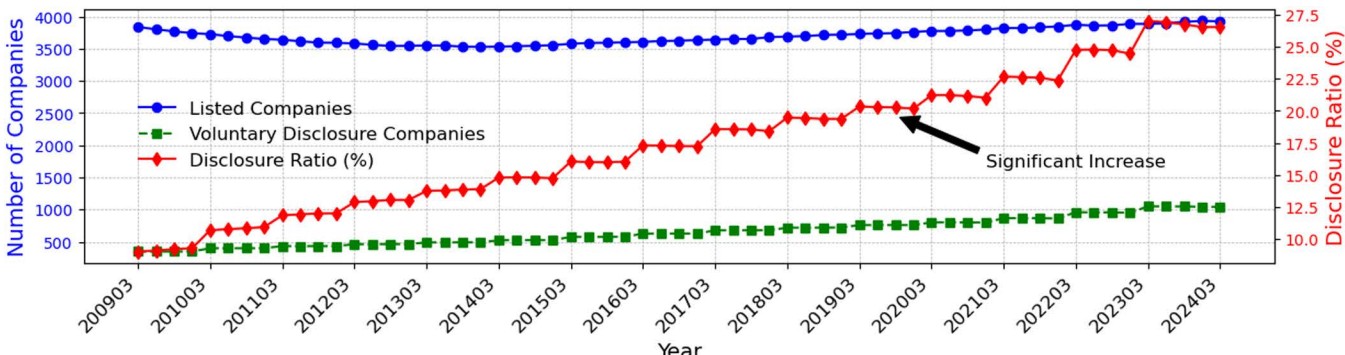

**Fig 2. Trends in listed and voluntary disclosure companies in Japan (2009-2024).** This figure shows the trend in the total number of listed companies in Japan and the number of companies engaging in voluntary disclosure. The right axis represents the percentage of companies practicing voluntary disclosure, displaying a consistent upward trend over the period.

listed market, industry classification, and shareholder composition, in order to gain a comprehensive understanding of how companies fulfill their transparency and social responsibility. Furthermore, based on these factors, the following hypotheses will be formulated.

The following hypotheses were developed for this study in order to identify the factors that influence voluntary disclosure by Companies. These hypotheses examine how financial indicators, Company characteristics, listing market, industry classification, and shareholder composition affect voluntary disclosure and are in a form that allows for testing whether the results are positive, negative, or irrelevant.

**2.2.1. Hypothesis 1: The impact of financial measures on voluntary disclosure.** A number of prior studies have suggested that financial indicators affect voluntary disclosure by Companies. It has been suggested that Companies with higher market capitalization reduce the cost of capital by disclosing integrated information, suggesting that stock market capitalization affects voluntary disclosure [15]. In addition to reports confirming that Companies with larger total assets disclose more information [16], there are also indications that larger Companies tend to report more detail and expand the scope of disclosure than smaller Companies [17]. Therefore, capitalization and the number of shares issued may also affect voluntary disclosure. Sales (Sales) and Operating Profit (Operating Profit), as indicators of company size, are expected to positively influence voluntary disclosure [18], but it has been pointed out that highly competitive companies, especially those with high profit margins, may not disclose information on sales and costs [19], so the impact may be both positive and it could occur passively. While dividend per share (DPS) indicates that dividends tend to increase in a transparent environment, DPS is expected to have an impact on voluntary disclosure [20], as there are indications that mandatory CSR disclosure will significantly reduce the amount of dividends paid by companies [21]. Finally, the price-to-book ratio (PBR) and price-to-earnings ratio (PER), which are ratios of the financial indicators presented above, are thought to influence disclosure behavior as indicators of a company's market valuation. A company with a high PER indicates high growth expectations, which may discourage voluntary disclosure, while a company with a low PER indicates a company with high growth expectations, which may discourage voluntary disclosure, PBR ratio reflects the value of a company's assets and is considered to be a factor that promotes disclosure.

Based on these studies, we believe that clarifying the impact of financial indicators on voluntary disclosure by Companies will help us understand how Companies strategically disclose information. Therefore, hypothesis H1 expects a meaningful relationship to exist between these financial indicators and voluntary disclosure reporting.

These arguments are supported by prior research emphasizing that voluntary disclosure, from an agency theory perspective, reduces information asymmetry and enhances transparency, thereby building trust with investors and regulators

[6]. Conversely, excessive disclosure can pose competitive risks, requiring companies to strategically balance transparency and competitive advantage [7].

**H1: Financial indicators influence the degree of voluntary disclosure.**

**2.2.2. Hypothesis 2: The impact of corporate characteristics on voluntary disclosure.** A number of prior studies have also suggested that corporate characteristics influence corporate voluntary disclosure. In addition to the suggestion that companies with a larger number of employees tend to disclose information about employee training more proactively [22], there are also indications that larger companies are more likely to disclose social information [23], suggesting that the number of employees has a positive impact on voluntary disclosure. Regarding corporate age, several studies [24,25] found that corporate age does not have a significant impact on voluntary disclosure, although more experienced Companies are likely to disclose more information to enhance their reputation and image. It will be interesting to see if the same results are obtained in other countries, as Japan has a large number of Companies with extremely high corporate age. On the other hand, it is reported that the number of years listed has no significant effect on disclosure in Indonesia [26], and it is possible that the length of listing also has a limited effect on voluntary disclosure. Regarding the adoption of IFRS (International Financial Reporting Standards), there are indications that IFRS may correct information deficiencies in financial statements and influence the disclosure behavior of companies [27]; there are indications that the adoption of IFRS will improve the quality and comparability of financial statements and increase the profitability of companies [28]; and the choice of GAAP is also expected to impact voluntary disclosure. With regard to listing in foreign markets, it has been shown that companies choose foreign stock markets based on signaling models and that the market environment may influence disclosure [29]. It has also been suggested that companies listed in multiple countries tend to disclose more ESG data than companies listed only in their home market in order to meet the demands of external capital markets [30]. There are also reports that corporate accounting disclosures are affected by investor demand for information, location, size, and listing on foreign markets [31]. There are also reports that listing on a foreign market affects disclosure behavior in response to the internationalization of legal systems and markets [32], and these factors may affect voluntary disclosure.

In addition, there are indications that share repurchases may intentionally increase unwelcome news because they lower the purchase price at which management purchases shares [33]. There is also a report that says that high-value companies buy back their own shares without disclosing them, increasing the liquidity of stock transactions and consequently reducing the equilibrium disclosure area [34], so we consider share buybacks as a key factor that affects the voluntary disclosure behavior of Companies. Finally, with regard to ISO certification, while reporting that ISO certification has no direct impact on carbon emissions disclosure, profitability and Company size have a positive impact on disclosure [35]. There is also a suggestion that ISO 14001 certification has no significant impact on greenhouse gas emissions disclosure [36], but Japan ISO certification is an important Explanation variable to understand the impact on voluntary disclosure behavior of Japanese companies.

Based on these studies, we believe that identifying the impact of Company characteristics on voluntary disclosure will help us understand how Companies formulate their disclosure strategies. Therefore, under Hypothesis H2, we expect a meaningful relationship to exist between corporate characteristics and voluntary disclosure.

These arguments are supported by previous research indicating that corporate disclosure is strategically influenced by the company's environment and market expectations [9] and that voluntary disclosures, especially integrated reports including sustainability and governance information, enhance decision usefulness and stakeholder relationship management [11].

**H2: Corporate characteristics determine the extent of voluntary disclosure.**

**2.2.3. Hypothesis 3: The impact of listing market factors on voluntary disclosure.** Numerous prior studies have provided insights into how the market characteristics on which a company is listed impact its voluntary disclosure. For

instance, stricter regulations in listed markets are associated with more active voluntary disclosure by companies aiming to enhance transparency [37,38]. Additionally, previous research suggests that stringent regulatory environments directly [39] influence corporate disclosure behaviors, creating incentives to disclose more comprehensive information. Fig 3 clearly demonstrates that voluntary disclosure in Japan between 2009 and 2024 has predominantly been carried out by companies listed in the Prime market segment.

This suggests that the prime market's strict regulations and high transparency requirements from investors may have a strong influence on companies' disclosure behavior. Companies listed on the standard, growth, and regional markets are expected to be more reluctant to disclose information because they are less regulated than those on the prime market.

Thus, it is important in this study to examine how listed markets affect voluntary disclosure by Companies. By analyzing the impact of specific market segments (prime market, standard market, growth market, and local market) on voluntary disclosure in Japan, we expect to clarify the role of market segments on corporate disclosure behavior.

This hypothesis is grounded in previous studies suggesting that companies strategically decide on disclosure depending on their competitive environment and market expectations [9] and that companies enhance transparency through voluntary disclosures to increase investor trust and meet market expectations [10].

**H3: The listing market determines the extent of voluntary disclosure.**

**2.2.4. Hypothesis 4: The impact of industry classification on voluntary disclosure.** Many studies have demonstrated that industry-specific characteristics significantly influence corporate voluntary disclosure behaviors, emphasizing that the regulatory and competitive environment of each industry impacts both the scope and content of disclosures [40,41]. Notably, research has reported particularly active voluntary disclosure practices within manufacturing industries, likely due to their higher environmental impact and consequent pressures for transparency [42]. Fig 4 clearly highlights trends in voluntary disclosure among Japanese companies across major industry categories from 2009 to 2024, illustrating that manufacturing significantly dominates voluntary disclosure activities. This suggests that industry classification plays a crucial role in determining corporate disclosure strategies.

Commercial (+101), Transport, Information and communications (+87), Finance and Insurance (+74), and Services (+66) also show a certain upward trend, while Mining (+1), Fishery, Agriculture & Forestry (+2), and Manufacturing (+3) are also on the rise. Mining (+1), Fishery, Agriculture & Forestry (+3), Electricity, Gas (+7), Real Estate (+15), and Construction (+35) have shown minor change in response to voluntary disclosure. Since these industries have different practices, it is important for this study to examine the impact of these differences. In particular, it is expected to clarify the role

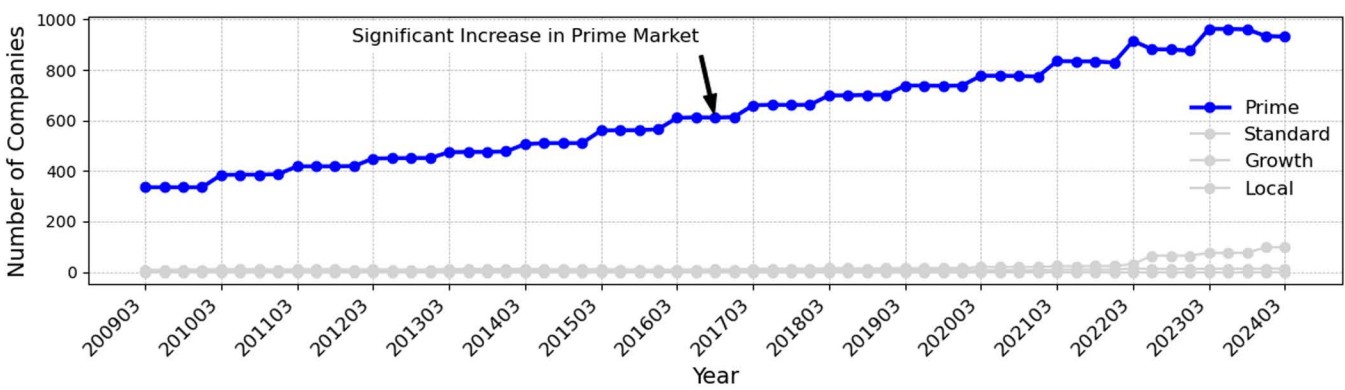

**Fig 3. Trends of voluntary disclosure by market segment in Japan (2009-2024).** This figure shows the trends of voluntary disclosure by market segment in Japan for the period from 2009 to 2024, highlighting the significant increase in the number of companies engaging in voluntary disclosure within the Prime market segment.

of industry characteristics on disclosure strategies by analyzing how industry classification affects voluntary disclosure in Japan.

This hypothesis draws upon earlier research indicating that companies strategically determine their disclosure behavior in response to specific regulatory pressures and competitive environments and that industry-specific characteristics critically shape corporate voluntary disclosure strategies [7].

**H4: Industry classification determines the extent of voluntary disclosure.**

**2.2.5. Hypothesis 5: The impact of shareholder composition on voluntary disclosure.** A number of previous studies have suggested the impact of shareholder composition on voluntary corporate disclosure. As so-called institutional investors, companies with a higher percentage of ownership by financial institutions (life insurance companies, non-life insurance companies, trust banks, ordinary banks, pension funds, etc.) tend to disclose information more proactively in order to increase transparency [43]. In addition, companies with major shareholders promote disclosure along with the introduction of independent directors [44]. Furthermore, it has been noted that sustainability-related disclosures have a positive and meaningful relationship with foreign and institutional shareholder composition and board independence [45]. Thus, shareholder composition has attracted attention as a factor that influences corporate disclosure behavior. In particular, companies in which financial institutions are major shareholders may promote voluntary disclosure because of the strong external demand for transparency. On the other hand, companies with many other corporations or individual shareholders are expected to face less pressure to disclose information and tend to restrain disclosure.

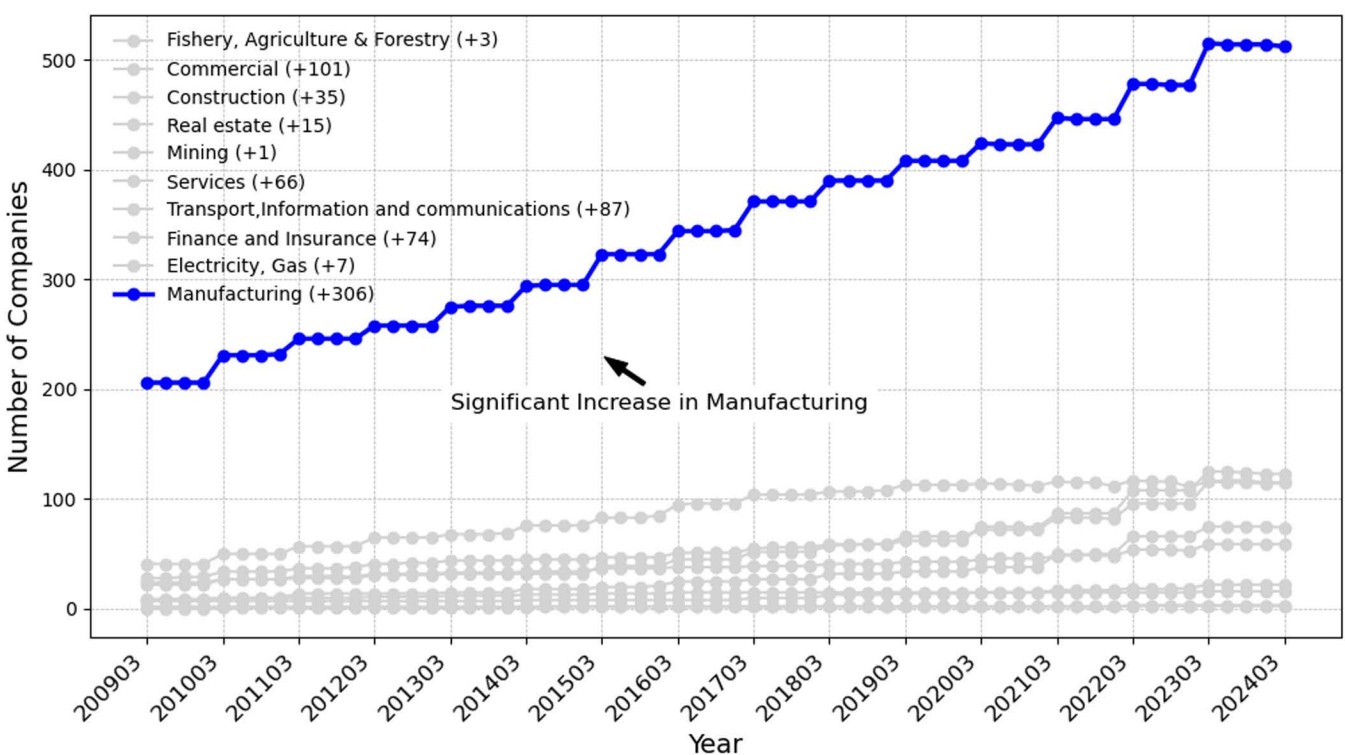

**Fig 4. Trends of voluntary disclosure by major industry categories in Japan (2009-2024).** This figure shows the trend of voluntary disclosure across major industries in Japan, highlighting a significant increase in the manufacturing sector, which has the highest number of companies engaged in voluntary disclosure from 2009 to 2024. The legend indicates the increase in the number of companies practicing voluntary disclosure from March 2009 to March 2024 for each industry.

Analyzing the impact of shareholder composition on corporate voluntary disclosure is important for understanding how shareholder type and composition affect disclosure strategies. Based on these studies, hypothesis H5 expects a significant relationship to exist between shareholder composition and the extent of voluntary disclosure.

This hypothesis is supported by prior research indicating that corporate disclosure strategies are significantly influenced by their competitive environment and strategic considerations related to maintaining competitive advantages, which vary according to shareholder composition and corporate governance structures [7].

**H5: Shareholder composition determines the extent of voluntary disclosure.**

Based on these previous studies, this study examines the impact of corporate financial indicators, corporate characteristics, listing market, industry classification, and shareholder composition on the voluntary disclosure behavior of Japanese Companies.

## 3. Research methods

### 3.1. Subject of research

The target of this study is all Companies listed on the Tokyo Stock Exchange and regional securities markets in Japan. This enables a comprehensive analysis of all listed companies in Japan as a whole.

### 3.2. Samples used

The analysis period of this study is the quarterly data from March 2009 to March 2024, and the total number of listed companies during this period is 5,915. The reason for using 2009 as the starting point is that the establishment of the International Integrated Reporting Council (IIRC) was proposed by the Prince of Wales in December of the same year [46]. This report analyzes what changes have occurred in the disclosure of information by Japanese companies since the proposal of this integrated reporting concept. The data (the table before regression) for reproducing the results of this experiment is provided as Supporting information (S1 File).

### 3.3. Research model

The research model utilized in this study is outlined in Fig 5, which depicts the conceptual framework along with the key variables employed to analyze determinants of voluntary disclosure. A detailed explanation of each variable and related information is provided in the subsequent sections.

### 3.4. Variable

**3.4.1. Object variable.** The Object variable in this study is Voluntary Reporting (VR), which indicates whether a Company prepares a voluntary disclosure report. It is defined as a dummy variable that is 0 if the Company does not disclose a voluntary disclosure report and 1 if it does. We collected PDF data by examining each Company's website, financial reports, and other publicly available information to see if the Company had a voluntary disclosure report. Although the name of the report may change from year to year, this study included in the analysis whether or not a report containing financial and non-financial information was issued, regardless of the name shown in Fig 1.

**3.4.2. Explanation variable.** Explanation variables are set (60 in total) in five categories of factors that affect voluntary disclosure. Detailed definitions of these variables and data processing methods (e.g., log transformations, dummy variable conversion, etc.) are provided in S2 File.

- H1: Nine items of β1_Financial Indicators (β1-1 to β1-9)

- H2: Eight items of β2_Company Characteristics (β2-1 to β2-8)

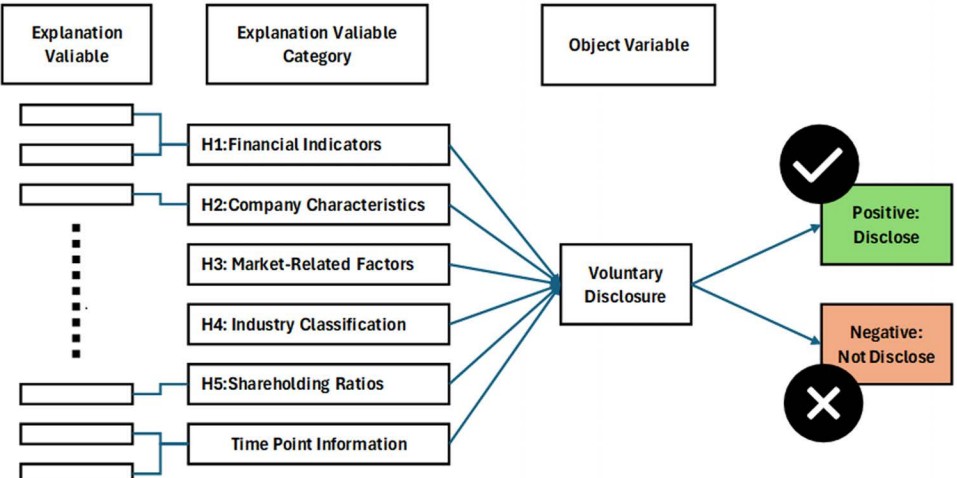

**Fig 5. Research framework.** This figure presents the conceptual framework illustrating the relationship between voluntary disclosure and its determinants, categorized into financial indicators, company characteristics, listed market segments, industry classification, shareholder composition, and the time dimension (quarter).

- H3: Four items of β3_Market-Related Factors (β3-1 to β3-4)

- H4: Thirty-three items of β4_Industry Classification (β4-1 to β4-33)

- H5: Five items of β5_Shareholding Ratios (β5-1 to β5-5)

- Point in time information (quarterly variable: β6_ Quarter)

### 3.5. Analysis model

**3.5.1. Model expression.** Binomial logistic regression analysis is employed to predict whether or not a company will prepare a voluntary disclosure report. This technique is appropriate when the objective variable takes on a binary value (0 or 1) and allows for a probabilistic estimation of the impact of the Explanation variable on the Objective variable. By using a logistic function, probabilities between 0 and 1 can be predicted from the weighted sum of the Explanation variables. In this study, we analyze how financial indicators, Company characteristics, listing market, industry classification, shareholder composition, and point in time information (quarterly) affect the voluntary disclosure behavior of Companies. The present model is expressed as follows:

$$logit\left(P\left(VR=1\right)\right)=\ \beta_0+\sum_{i=1}^{9}\beta1-i{\cdot}x_{1-i}+\sum_{J=1}^{8}\beta2-j\cdot x_{2-j}+\sum_{k=1}^{4}\beta3-k\cdot x_{3-k}+\sum_{l=1}^{33}\beta4-l\cdot x_{4-l}+\sum_{m=1}^{5}\beta5-m{\cdot}x_{5-m}+\beta6\cdot Quarter+\ \in$$

Here, VR is the Object variable and is set to 1 if the company issues voluntary disclosure reports and 0 if it does not. In addition, the Explanation variable (See S2 File: Explanation Variables)) falls into the five categories described above.

**3.5.2. Analytical method.** *3.5.2.1. Binomial logistic regression*: In this study, a binomial logistic regression was used to predict whether a Company would issue a voluntary disclosure report. This method is appropriate when the Object variable is binary (0 or 1) and shows the effect of a change in the Explanation variable on the probability of occurrence. The estimated coefficients (β values) indicate the impact of the Explanation variable on the probability of any disclosure. Specifically, it is interpreted as follows:

- The regression coefficient (Coef) represents the logarithm of the odds ratio for a one-unit increase in the Explanation variable.

- Odds (Odds) is the ratio of the probability that a company will make a voluntary disclosure to the probability that it will not. If the odds ratio is greater than 1, the probability of voluntary disclosure increases.

$$Odds = \frac{P(VR=1)}{P(VR=0)}$$

- The probability is the probability that a company will make a voluntary disclosure. It is derived from the odds ratio by the following equation:

$$P(VR=1) = \frac{1}{1 + e^{-logit(P(VR=1))}}$$

*3.5.2.2. Visualization by sigmoid function*: To make the results of the logistic regression more intuitive, the influence of the Explanation variable was visualized using a sigmoid function. This function shows the variation in arbitrary disclosure probability due to changes in the Explanation variable as an S-shaped curve, which is classified into three main patterns

- **Rising right:** the probability of voluntary disclosure increases as the Explanation variable increases.

- **Rightward**: the probability of voluntary disclosure declines as the Explanation variable increases.

- **Other**: Explanation variable shows no clear effect on voluntary disclosure.

The sigmoid function is shaped by the positive and negative regression coefficients (Coef) and shows a sharp change in the middle part of the curve as the odds (Odds) change. This visualization provides a visual understanding of how the Explanation variable is affecting the voluntary disclosure.

*3.5.2.3. Coefficients, odds, and probability correlations*: In logistic regression, regression coefficients, odds, and probabilities are closely related. Therefore, how each Explanation variable affects the probability of voluntary disclosure is also analyzed in terms of regression coefficients (Coef) and odds (Odds). When the regression coefficients are positive, the odds exceed 1, the sigmoid curve rises steadily, and the probability of voluntary disclosure increases. When the regression coefficient is negative, the odds are less than 1, the sigmoid curve falls steadily, and the probability of voluntary disclosure decreases.

*3.5.2.4. Validation of analytical methods*: In this study, to avoid multicollinearity, the independence of each variable was verified using VIF (variance expansion factor), and the model was built to the extent that there were no problems. In addition, training and test data were fairly distributed by data partitioning to prevent overtraining of the model and ensure generality of the results.

*3.5.2.5. Response to multicollinearity*: Since a large number of Explanation variables are used in this model, there is concern about the occurrence of multicollinearity. To address this, the VIF (variance expansion factor) was calculated and multicollinearity was examined for Explanation variables with a VIF greater than ten. In particular, for β3_Market, β4_Sector, and β5_Shareholding Ratio, the Explanation variables with the highest frequency (e.g., β3-1(P): Prime Market, β4-6-25: Information and Communication Industry, β5-4: Shareholder Composition_Personal and Others) were used as reference categories and deducted before analysis. The VIF is deducted prior to the analysis. Other Explanation variables with VIF greater than 10 (e.g., total stock market capitalization, total assets, etc.) were retained as explanation variables after considering their content and determining that they were not multicollinear (see Table 3-1 in S3 File for details).

**3.5.3. Response to data splitting.** Data partitioning is important to avoid over-learning and to improve generalization ability in model construction and evaluation. In this study, data are split by Company ID to take into account the different characteristics of each Company. This prevents information from crossing across Companies and improves the generalization performance of the model.

Specifically, the following procedure was used to split the data.

1. Split by company ID: Group data by company ID and split them into training and test data. This method prevents data crossover between different companies and enables more accurate model evaluation.

2. Split training and test data: 80% of the total data is used as training data and the remaining 20% as test data. In this way, the model is trained on the training data and the performance of the model is evaluated on the test data.

3. Confirmation of the balance of the split: Check that the number of Companies and data are distributed in a balanced manner, including industry sectors. This ensures unbiased data partitioning and fair model evaluation.

## 4. Empirical results

### 4.1. Overall model evaluation

The results of the logistic regression analysis confirm that the model is fully accurate. The overall correctness rate (Accuracy) is 89.18%, which accurately predicts the presence of voluntary disclosure (see Table 3-1 in S3 File). The AUC (area under the ROC curve) was as high as 0.9449, indicating excellent ability to distinguish between voluntary (VR = 1) and involuntary (VR = 0) disclosures (see Table 3-4 in S3 File).

The Confusion Matrix results show that 33,826 "involuntary disclosures (VR=0)" were correctly predicted with 96% accuracy, while 5,840 "voluntary disclosures (VR=1)" were correctly predicted with 64% accuracy, and 3,326 were incorrectly labeled "involuntary disclosures" (see Table 3-2 in S3 File). Furthermore, according to the Classification Report, the compliance rate for "Non-Voluntary Disclosure (VR=0)" was 0.91, the reproducibility rate was 0.96, and the F1 score was extremely high at 0.93. In contrast, for "Voluntary Disclosure (VR=1)," the fit rate is 0.80, the repeatability is 0.64, and the F1 score is 0.71, which are slightly lower, but still retain adequate performance (see Table 3-3 in S3 File).

### 4.2. Influence analysis of Explanation variable

The coefficients, standard errors, and p-values of the explanatory variables were analyzed to test both the significance and direction of their impact on voluntary disclosure (detailed results are provided in S4 File). As a result, a clear distinction emerged among hypotheses concerning variables that positively or negatively influenced voluntary disclosure. Fig 6 illustrates these logistic regression results, clearly categorizing each explanatory variable according to its corresponding hypothesis (H1–H5) and highlighting positive or negative influences.

### 4.3. Results of visualization by sigmoid function

To visually understand how explanatory variables affect voluntary disclosure behavior, we visualized the results using a sigmoid function, as shown in Fig 7 (illustrating the relationship between explanatory variables and the probability of voluntary disclosure via logistic curves).

• A steadily rising sigmoid function indicates that the probability of voluntary disclosure increases as the Explanation variable increases, indicating that Companies tend to disclose information proactively. For example, market value and number of employees fall into this category.

• A steadily declining sigmoid function indicates that the probability of voluntary disclosure decreases as the Explanation variable increases, indicating a tendency for Companies to restrain disclosure; this is the case for PER and foreign market listings.

• Other sigmoid functions represent cases where Explanation variables do not show a clear impact on voluntary disclosures. Operating income and dividends fall into this category.

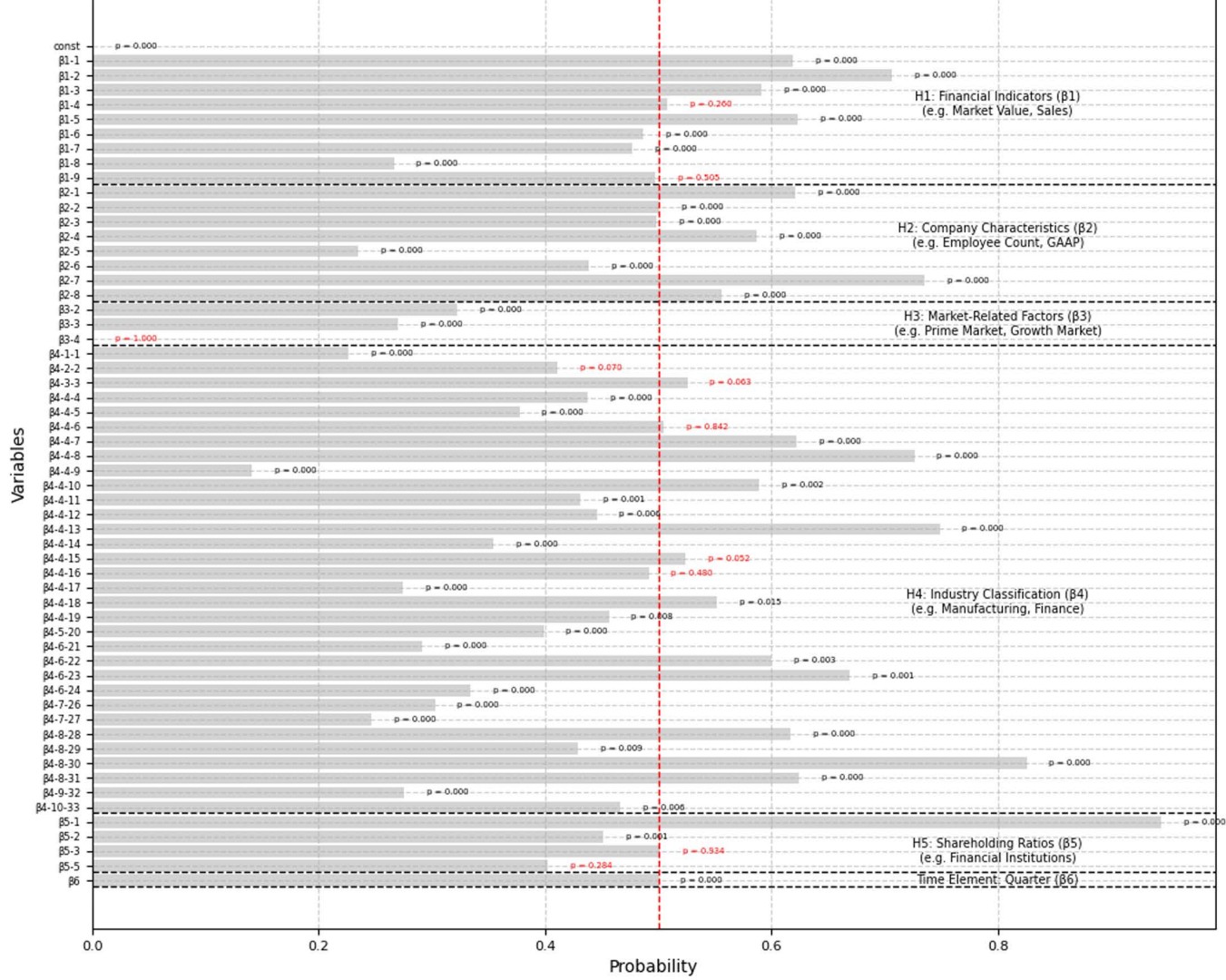

**Fig 6. Logistic regression coefficients grouped by hypothesis.** This figure presents logistic regression results grouped by the corresponding hypotheses: H1 (Financial Indicators), H2 (Company Characteristics), H3 (Market-Related Factors), H4 (Industry Classification), H5 (Shareholding Ratios), and an additional time element (Quarter). Dashed lines separate each hypothesis clearly. The objective variable analyzed is whether a company engages in voluntary disclosure or not.

These results show that different Explanation variables have different effects on voluntary disclosure behavior and clearly indicate how each variable acts on voluntary disclosure based on the form of the sigmoid function. The following sections summarize the results of the analysis for each hypothesis and clarify points to be considered in the discussion.

**4.3.1. Hypothesis H1: The impact of Financial Indicators on voluntary disclosure.** Market value (β1-1), total assets (β1-2), and capital stock (β1-3) were found to increase the likelihood of voluntary disclosure as initially hypothesized. This relationship is clearly illustrated in Fig 8, which demonstrates the positive influence of these financial indicators on the probability of voluntary disclosure.

These results are consistent with previous studies [15,16] that larger Companies are more active in disclosure. We also confirm that sales (β1-5), contrary to expectations, promote voluntary disclosure. Operating profit (β1-6) and PBR (β1-7)

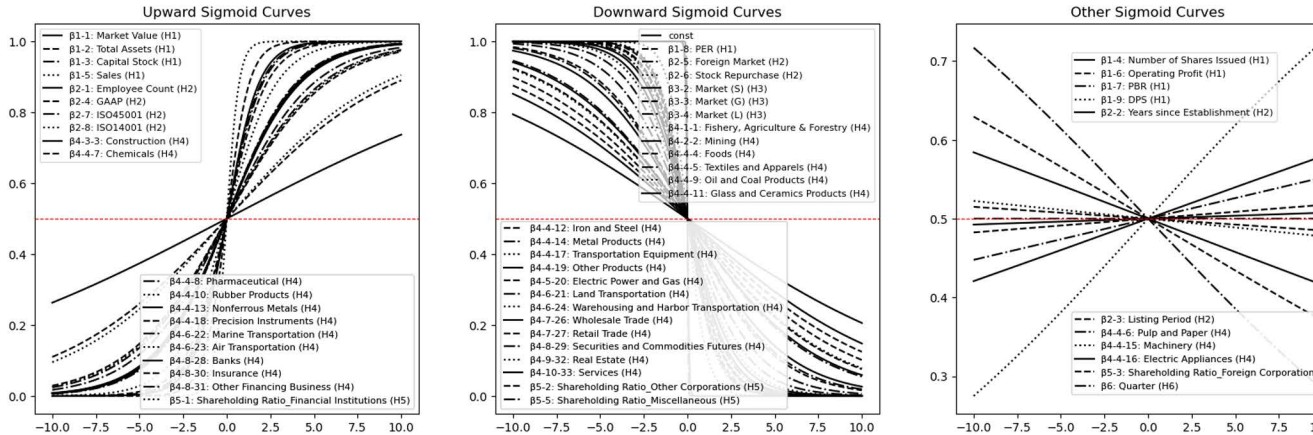

**Fig 7. Combined upward, downward, and other sigmoid curves.** This figure combines three types of sigmoid curves: upward, indicating a positive relationship; downward, indicating a negative relationship; and other, indicating minimal or no relationship with voluntary disclosure.

were statistically significant but had limited impact on voluntary disclosure, while PER (β1-8) confirmed a "right-sided" relationship that suppressed voluntary disclosure, indicating that Companies with higher price-to-earnings ratios tend to suppress information disclosure. As a new finding, this result indicates that Companies with higher PERs may suppress disclosure; no statistically significant effects were found for ISSUEDSHARE (β1-4) or DPS (β1-9). Detailed results are presented in S5 File.

**4.3.2. Hypothesis H2: The impact of Company Characteristics on voluntary disclosure.** Employee count (β2-1), GAAP(β2-4), ISO 45001 certification (β2-7), and ISO 14001 certification (β2-8) positively influenced voluntary disclosure, which aligns with findings from prior studies [25,26,31]. These relationships are clearly depicted in Fig 9, highlighting that these corporate characteristics significantly enhance the likelihood of voluntary disclosure. Conversely, foreign market listing (β2-5) and stock repurchases (β2-6) were found to negatively affect voluntary disclosure, contradicting earlier studies which indicated that foreign market listings typically promote disclosure. Additionally, variables such as years since establishment (β2-2) and listing period (β2-3), though statistically significant, showed no substantial impact on voluntary disclosure behaviors. Detailed statistical results are provided in S5 File.

**4.3.3. Hypothesis H3: The impact of Market-Related Factors on voluntary disclosure.** Hypothesis H3 examined the impact of the listing market on companies' voluntary disclosure. As illustrated in Fig 10, companies listed on the Standard (β3-2) and Growth (β3-3) markets are less likely to engage in voluntary disclosure compared to those listed on the Prime market. This finding aligns with prior studies, suggesting that stricter market regulations correlate with greater transparency and voluntary disclosure activities.

These results confirm that the more regulated the market, the more transparent the disclosure with a previous study [38]. For local markets (β3-4), no statistically significant effects were identified, suggesting that there is no significant impact on voluntary disclosure. Detailed results are provided in S5 File.

**4.3.4. Hypothesis H4: The impact of Industry Classification on voluntary disclosure.** Hypothesis H4 provides an analysis of how industry classification influences voluntary disclosure among companies across various sectors. As Fig 11 illustrates, there are notable differences in voluntary disclosure practices across industry categories. Positive relationships are identified in industries including Insurance, Nonferrous Metals, Pharmaceuticals, Air Transportation, and Chemicals, indicating that these sectors tend to actively disclose information voluntarily. In contrast, industries such as Oil, Agriculture, and Retail exhibit negative relationships, suggesting a tendency to avoid voluntary disclosure.

Due to the wide variety of industries, we adjusted S5 File to examine each industry and prepared the following Table 2.

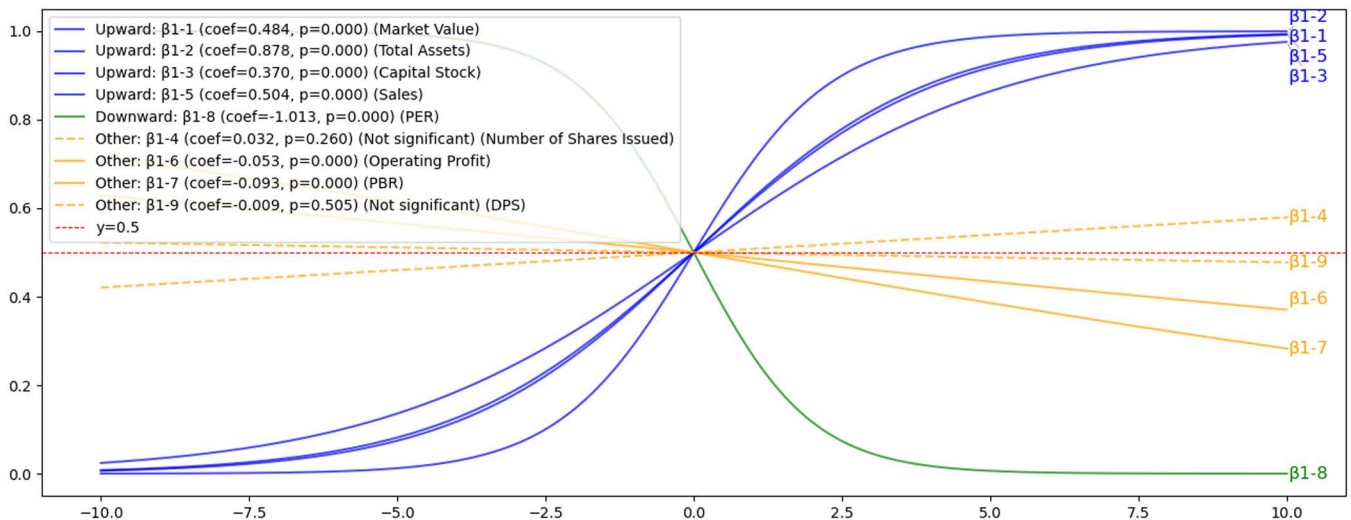

**Fig 8. H1: Results of sigmoid function for the impact of financial indicators on voluntary disclosure.** This figure illustrates how financial indicators influence voluntary disclosure behavior using sigmoid curves. The results indicate a positive relationship between voluntary disclosure probability and financial measures such as Market Value, Total Assets (TTLASSET), Capital Stock, and Sales. Conversely, the Price-to-Earnings Ratio (PER) demonstrates a negative relationship. Number of Issued Shares, Operating Profit, Price-to-Book Ratio (PBR), and Dividends per Share (DPS) do not exhibit significant effects.

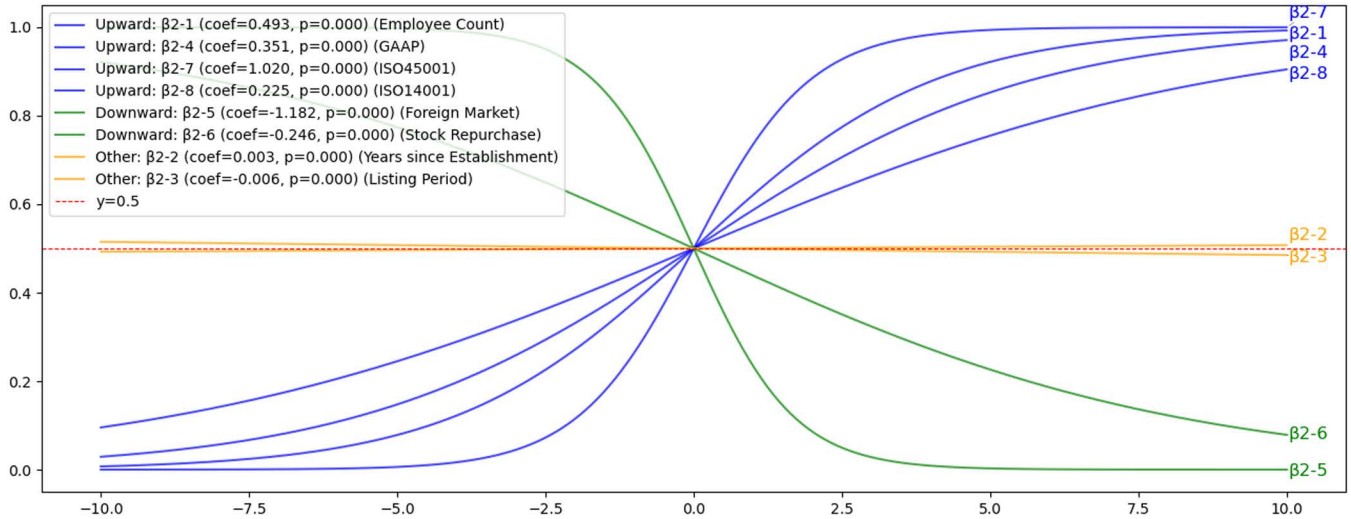

**Fig 9. H2: Results of sigmoid function for the influence of corporate characteristics on voluntary disclosure.** This figure demonstrates the influence of various company characteristics on voluntary disclosure probability. Positive relationships are observed with Employee Count, GAAP compliance, ISO 45001 certification, and ISO 14001 certification, indicating that increases in these variables correlate with greater likelihood of voluntary disclosure. In contrast, foreign market listings and stock repurchase activities show negative relationships. Variables such as years since establishment and listing period show minimal or no significant impact.

Below are the results for major industries.

1. **Fishery, Agriculture & Forestry: new findings: none**

One medium classification included in Fishery, Agriculture & Forestry confirmed the negative impact of voluntary disclosure as predicted.

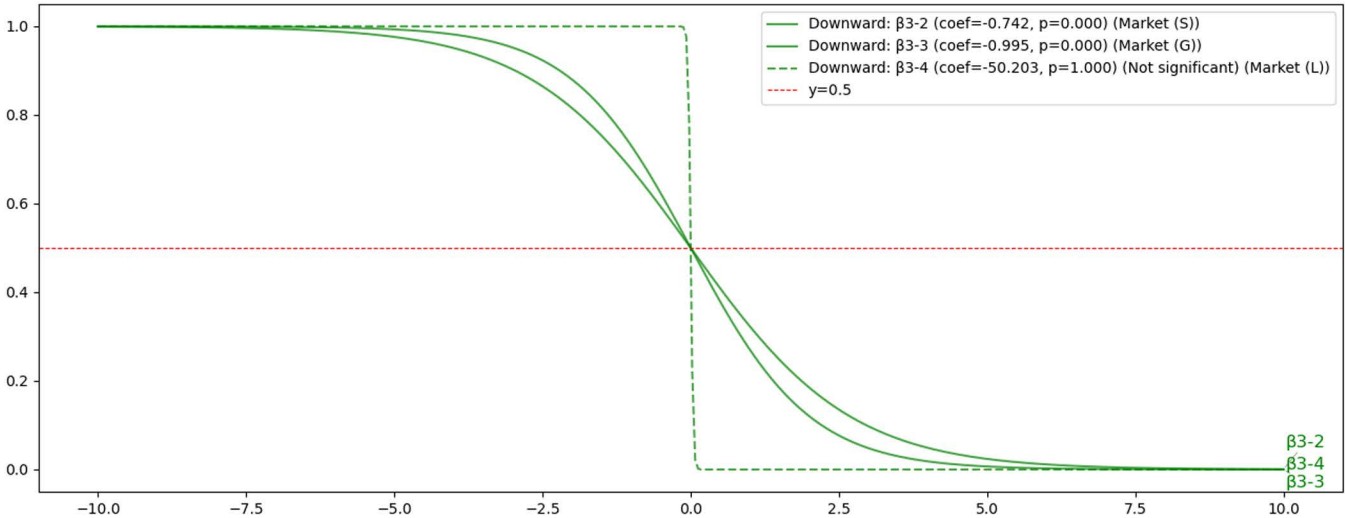

**Fig 10. H3: Results of Sigmoid Function for the Impact of Listing Market Factors on Voluntary Disclosure.** This figure demonstrates the relationship between listing market categories and the probability of voluntary disclosure. The sigmoid curves indicate that companies listed on the Standard, Growth, and Local market segments are relatively less likely to voluntarily disclose information than those listed on the Prime market. Notably, the Local market does not exhibit statistically significant effects.

2. **Mining: new findings: none**

No statistically significant results were obtained for the one medium classification included in Mining.

3. **Construction: new findings: none**

No statistically significant results were obtained for the one medium classification included in Construction.

4. **Manufacturing: new findings: including**

The largest number of medium-sized industries included in Manufacturing was 16, of which there were three that were not statistically favorable. Although our initial expectation was that the respondents would be more willing to voluntarily disclose, only five industries (β4-4-7: Chemicals, β4-4-8: Pharmaceuticals, β4-4-10: Rubber Products, β4-4-13: Nonferrous Metals, and β4-4-18: Precision Instruments) were more willing to do so, while eight industries were less willing. and β4-4-18: Precision Instruments), and only eight industries (β4-4-4: Foods, β4-4-5: Textiles and Apparels, β4-4-9: Oil and Coal Products, β4-4-11: Glass and Ceramics Products, β4-4-12: Glass and Ceramic Products, and β4-4-13: Precision Instruments). Ceramics Products, β4-4-12: Iron and Steel, β4-4-14: Metal Products, β4-4-17: Transportation Equipment, and β4-4-19: Other Products), and the results were negative. The results of the survey revealed new findings that were not apparent from the increase in the overall disclosure rate alone.

5. **Electricity, Gas: new findings: none**

One medium classification included in Electricity, Gas Fishery, Agriculture & Forestry confirmed the negative impact of voluntary disclosure as predicted.

6. **Transport, Information and Communications: new findings: including**

There are five subcategories in the Transport, Information and Communications category, and for the four subcategories other than the reference category, two industries (B 4-6-22: Marine Transportation and B 4-6-23: Air Transportation) were proactive in voluntary disclosure, as expected, while the remaining two industries (B 4-6-21: Land Transportation and B

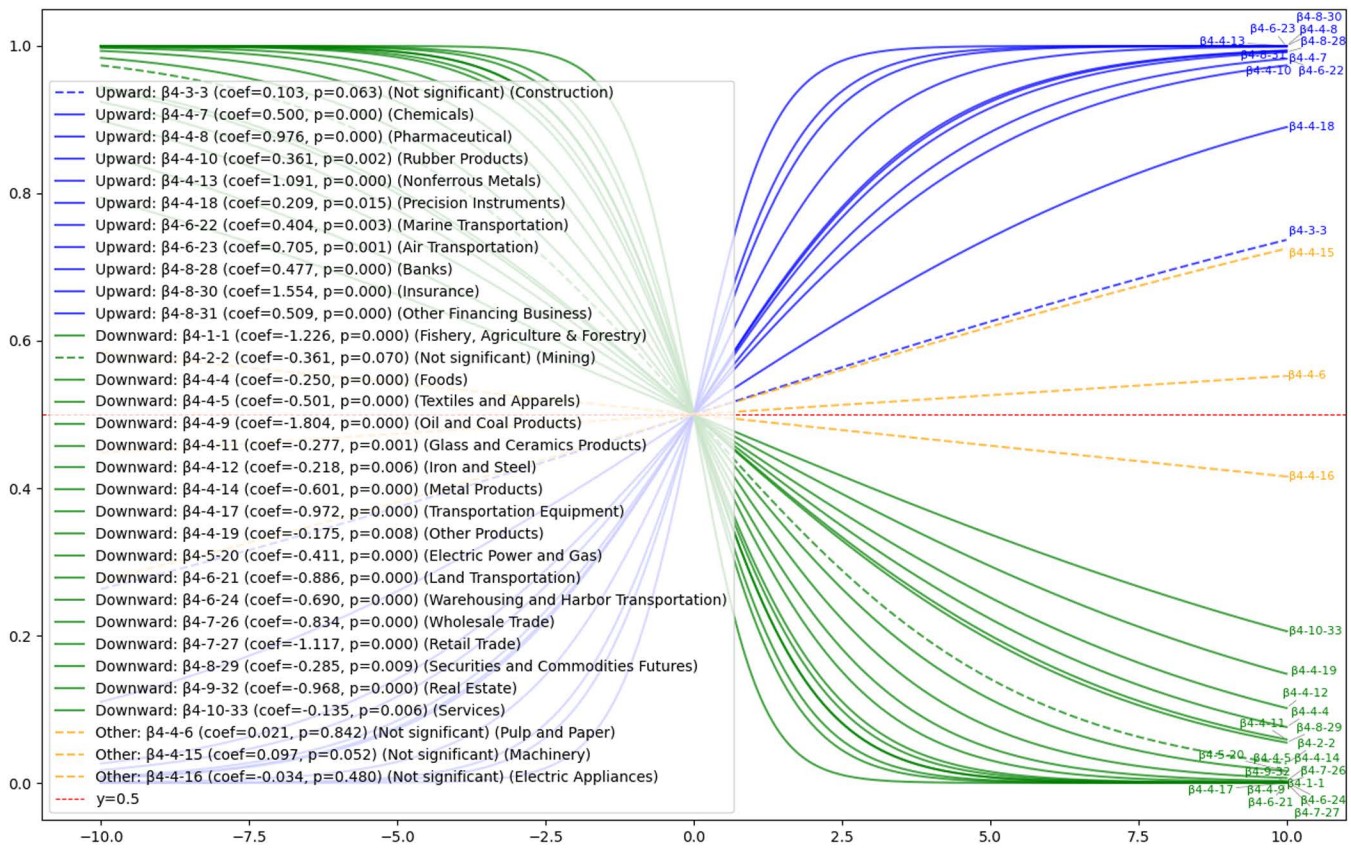

**Fig 11. H4: Results of sigmoid function for the effect of industry classification on voluntary disclosure.** This figure illustrates industry-specific variations in voluntary disclosure probability. Industries like Insurance, Nonferrous Metals, Pharmaceuticals, Air Transportation, and Chemicals display positive associations with voluntary disclosure, whereas industries such as Oil, Agriculture, and Retail show negative associations. Several other industries, including Construction and Machinery, show no significant influence on voluntary disclosure behaviors.

4-6-24: Warehousing and Harbor Transportation) were reactive. Air Transportation), while the remaining two (B 4-6-21: Land Transportation and B 4-6-24: Warehousing and Harbor Transportation) were not. The results of the two industries (β4-6-21: Land Transportation and β4-6-24: Warehousing and Harbor Transportation) showed a negative trend.

## 7. Commercial: new findings: including

Although we had expected positive effects on voluntary disclosure in two of the medium categories included in the Commercial category (β4-7-26: Wholesale Trade and β4-7-27: Retail Trade), we found negative effects on voluntary disclosure in both categories. This is a different result from other industries, where there is a mixture of positive and negative voluntary disclosure, and a new finding was detected that could not be determined by the increasing trend in the major categories.

## 8. Finance and Insurance: new findings: including

Finance and Insurance included four subcategories, three of which (β4-8-28: Banks, β4-8-30: Insurance, and β4-8-31: Other Financing Business) were active in voluntary disclosure, as expected, while the remaining one (β4-8-29: Securities and Commodities Futures) was passive. The remaining one (β4-8-29: Securities and Commodities Futures) was negative, revealing new insights that could not be gleaned from the general classification of increasing trends.

Table 2. Results of Hypothesis H4—Impact of industry classification on voluntary disclosure.

| Reference | 10 major industry categories | Assumed | 33 industry subcategories (Explanatory Variable) | Observed | Statistically Significant* /NS: Not statistically significant | Result | Unexpected Findings |
|---|---|---|---|---|---|---|---|
| B 4-1-1 | Fishery, Agriculture & Forestry | →↘ | Fishery, Agriculture & Forestry | ↘ | * | ↘ | – |
| B 4-2-2 | Mining | →↘ | Mining | ↘ | NS | – | – |
| B 4-3-3 | Construction | →↘ | Construction | ↗ | NS | – | – |
| B 4-4-4 | Manufacturing | ↗ | Foods | ↘ | * | ↘ | Yes |
| B 4-4-5 | | ↗ | Textiles and Apparels | ↘ | * | ↘ | Yes |
| B 4-4-6 | | ↗ | Pulp and Paper | → | NS | – | – |
| B 4-4-7 | | ↗ | Chemicals | ↗ | * | ↗ | – |
| B 4-4-8 | | ↗ | Pharmaceutical | ↗ | * | ↗ | – |
| B 4-4-9 | | ↗ | Oil and Coal Products | ↘ | * | ↘ | Yes |
| B 4-4-10 | | ↗ | Rubber Products | ↗ | * | ↗ | – |
| B 4-4-11 | | ↗ | Glass and Ceramics Products | ↘ | * | ↘ | Yes |
| B 4-4-12 | | ↗ | Iron and Steel | ↘ | * | ↘ | Yes |
| B 4-4-13 | | ↗ | Nonferrous Metals | ↗ | * | ↗ | – |
| B 4-4-14 | | ↗ | Metal Products | ↘ | * | ↘ | Yes |
| B 4-4-15 | | ↗ | Machinery | → | NS | – | – |
| B 4-4-16 | | ↗ | Electric Appliances | → | NS | – | – |
| B 4-4-17 | | ↗ | Transportation Equipment | ↘ | * | ↘ | Yes |
| B 4-4-18 | | ↗ | Precision Instruments | ↗ | * | ↗ | – |
| B 4-4-19 | | ↗ | Other Products | ↘ | * | ↘ | Yes |
| B 4-5-20 | Electricity, Gas | →↘ | Electric Power and Gas | ↘ | * | ↘ | – |
| B 4-6-21 | Transport, Information and communications | ↗ | Land Transportation | ↘ | * | ↘ | Yes |
| B 4-6-22 | | ↗ | Marine Transportation | ↗ | * | ↗ | – |
| B 4-6-23 | | ↗ | Air Transportation | ↗ | * | ↗ | – |
| B 4-6-24 | | ↗ | Warehousing and Harbor Transportation | ↘ | * | ↘ | Yes |
| B 4-6-25 | | ↗ | Information & Communication | ↗ | Baseline | – | – |
| B 4-7-26 | Commercial | ↗ | Wholesale Trade | ↘ | * | ↘ | Yes |
| B 4-7-27 | | ↗ | Retail Trade | ↘ | * | ↘ | Yes |
| B 4-8-28 | Finance and Insurance | ↗ | Banks | ↗ | * | ↗ | – |
| B 4-8-29 | | ↗ | Securities and Commodities Futures | ↘ | * | ↘ | Yes |
| B 4-8-30 | | ↗ | insurance | ↗ | * | ↗ | – |
| B 4-8-31 | | ↗ | Other Financing Business | ↗ | * | ↗ | – |
| β 4-9-32 | Real Estate | →↘ | Real Estate | ↘ | * | ↘ | – |
| β 4-10-33 | Services | ↗ | Services | ↘ | * | ↘ | Yes |

9. **Real Estate: New Findings: None**

One medium classification included in Real Estate confirmed the negative impact of voluntary disclosure as predicted.

10. **Services: new findings: including**

Contrary to the initial prediction that one medium classification (β4-10-33: Services) included in Services would have a positive impact on voluntary disclosure, a negative impact was identified, which is considered a new finding.

Overall, the impact of industry classification on the voluntary disclosure behavior of Companies varies from industry to industry and may require consideration of the severity of regulations and horizontal practices in the industry. Explanation variables that were found to diverge from previous studies will be subject to additional consideration in the Discussion section, based on their peculiarities in Japan. Detailed results by industry will be provided in S5 File.

**4.3.5. Hypothesis H5: The impact of Shareholding Ratios on voluntary disclosure.** The impact of shareholder composition on voluntary disclosure is diverse. The shareholding by financial institutions (β5-1) promotes voluntary disclosure, consistent with a previous study [46], as illustrated in Fig 12. Conversely, a higher proportion of shareholding by other corporations (β5-2) is associated with a lower likelihood of disclosure, a finding that contrasts with earlier research and suggests strategic considerations to protect competitive advantages. Ownership by foreign corporations (β5-3) and miscellaneous individual shareholders (β5-5) showed no significant effects on voluntary disclosure. Detailed analytical results are provided in S5 File.

The analysis included a temporal factor (β6), represented by quarterly dummy variables, to investigate potential trends or seasonal effects on voluntary disclosure behavior. As illustrated in Fig 13, the findings suggest that the quarter (time-point information) did not significantly impact voluntary disclosure decisions by companies in this study. Thus, temporal factors did not exhibit any consistent or meaningful influence on disclosure behavior.

Although the quarter's point in time was replaced by a dummy variable for convenience, this study suggests that it has no effect on voluntary disclosure.

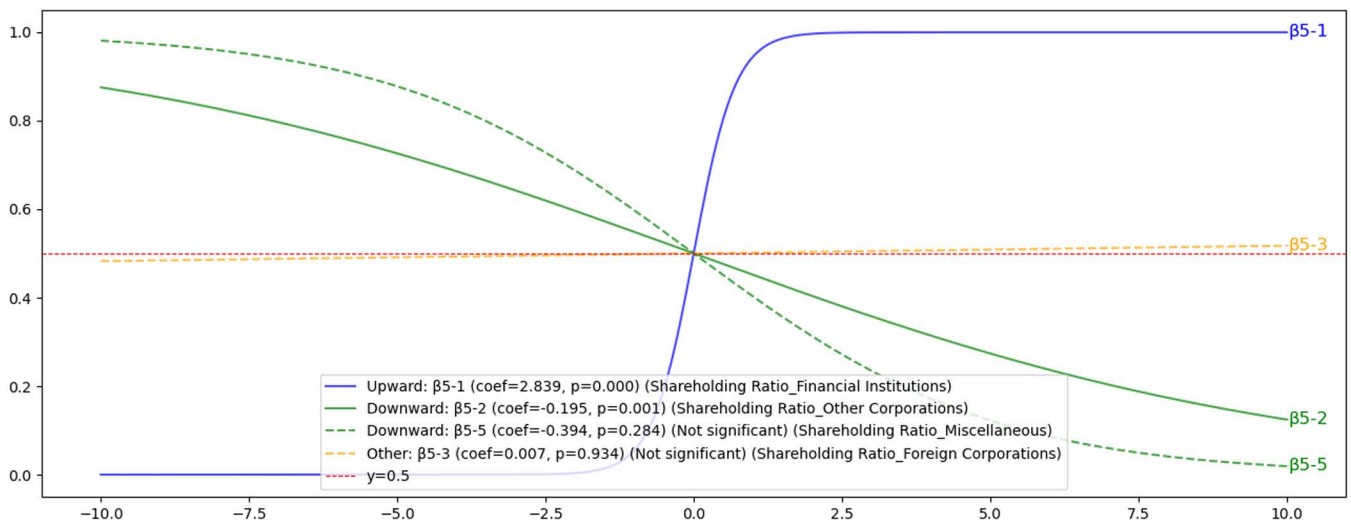

**Fig 12. H5: Results of sigmoid function for the influence of shareholder composition on voluntary disclosure.** This figure highlights the diverse influences of shareholder composition on voluntary disclosure behavior. Specifically, a higher shareholding ratio by financial institutions positively impacts voluntary disclosure, while ownership by other corporations negatively affects disclosure probability. No significant relationships were found for shareholding by miscellaneous entities or foreign corporations.

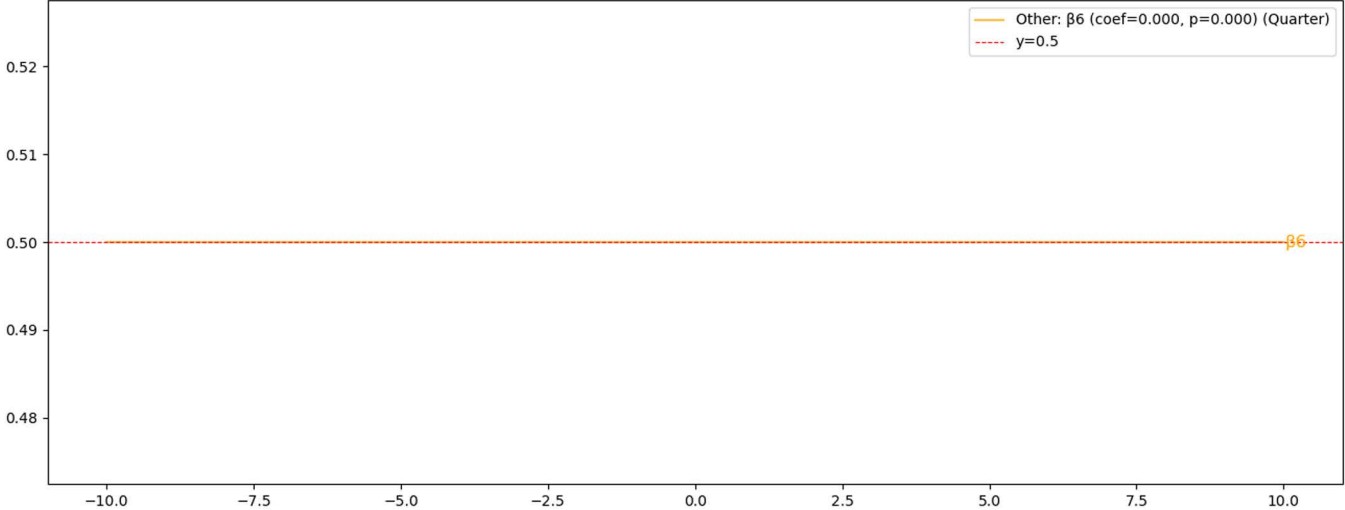

**Fig 13. Results of sigmoid function for the influence of time-point information on voluntary disclosure.** This figure illustrates the relationship between the quarterly time-point variables and voluntary disclosure behavior. The analysis indicates no statistically significant effect of these temporal dummy variables, suggesting that disclosure decisions are independent of specific quarterly timing within the period analyzed.

Based on the results of our previous analysis, we will proceed with a discussion of the impact of company size, financial indicators, company characteristics, listing market, industry classification, and shareholder composition on voluntary disclosure behavior. Detailed discussion will be based on the following points.

1. **Impact of Financial Indicators**: While a company's size and profitability act as drivers of voluntary disclosure, some indicators, such as PER, have a negative impact on disclosure. The implications of these results and why each indicator has a different impact will be discussed.

2. **Impact of Company Characteristics**: While the number of employees, adoption of GAAP, and acquisition of ISO promote voluntary disclosure, foreign market listings and share repurchases discourage voluntary disclosure. We will examine what factors are behind this trend.

3. **Impact of Listed Markets**: As originally planned, but discussion of the impact of different market regulations and expectations on voluntary disclosure will focus on the differences between prime and other markets.

4. **Impact of Industry Classification**: Certain industries are more active in voluntary disclosure, while others are more reluctant. This section examines how the characteristics of each industry affect disclosure behavior.

5. **Impact of shareholder composition**: Discuss why the shareholder composition of financial institutions promotes voluntary disclosure while other corporations discourage voluntary disclosure.

## 5. Discussion

### 5.1. On the development of model equations

In this study, we analyzed the factors affecting voluntary corporate disclosure by dividing them into five categories: financial indicators, corporate characteristics, listed market, industry classification, and shareholder composition. The initial model equation was expressed as follows:

$$logit\left(P\left(VR=1\right)\right) = \beta_0 + \sum_{i=1}^{9}\beta 1-i \cdot x_{1-i} + \sum_{J=1}^{8}\beta 2-j \cdot x_{2-j} + \sum_{k=1}^{4}\beta 3-k \cdot x_{3-k} + \sum_{l=1}^{33}\beta 4-l \cdot x_{4-l}$$
$$+ \sum_{m=1}^{5}\beta 5-m \cdot x_{5-m} + \beta 6 \cdot Quarter + \in$$

where each β corresponds to each Explanation variable and X represents the observed value of each variable.

However, based on the post-experimental results, it became clear that the Explanation variable could be categorized into three main sigmoidal forms: **rising right**, **falling right**, and other (flat or atypical shape). This indicated that the model equations can be aggregated as follows:

$$\text{logit}\left(P\left(VR=1\right)\right) = \beta0 + \beta_{upward} \cdot \text{Upward Factors} + \beta_{downward} \cdot \text{Downward Factors} + \beta_{other} \cdot \text{Other Factors} + \beta_{quarter} \cdot Quarter + \in$$

whereas:

- **Upward Factors** are factors that promote voluntary disclosure, such as Market Value, Total Assets, Capital Stock, Sales, Employee Count, GAAP, ISO 45001, ISO 14001, some of Industries and Shareholding Ratio_Finacial institutions.

- **Downward Factors** are factors that inhibit voluntary disclosure, such as PER, Foreign Market, Stock Repurchase, MARKET(S), MARKET(G), some of Industries and Shareholding Ratio_Other corporations.

- **Other Factors** are factors such as IssuedShareOperating Profit, PBR, Years since establishment, listing period, DPS and some of Industries are not statistically significant or have no clear impact.

This aggregation has resulted in a simple and effective restructuring of the original complex model, making it easier to visually understand the effects of each Explanation variable. Next, we proceed with a discussion of how these factors affect the voluntary disclosure behavior of companies, based on each hypothesis.

### 5.2. Discussion of each hypothesis

Table 3 summarizes the results of the analysis of each Explanation variable.

In the following, a discussion will be conducted on Explanation variables for which results that differ from predictions based on prior studies and other sources are detected.

**5.2.1. Hypothesis H1: The impact of financial indicators on voluntary disclosure.** With regard to financial indicators, β1-5 (Sales), β1-6 (Operating Profit), β1-7 (PBR), and β1-8 (PER) yielded results that differ from the forecasts. A discussion of each variable is presented below.

1. **β1-5: Sales**

Sales (β1-5), whose negative impact on voluntary disclosure had been verified in previous studies, was confirmed to have a positive impact in the present experimental results, on the contrary. As was observed in the trends of other Explanation

**Table 3. Summary of experimental results.**

| | H1 | H2 | H3 | H4 | H5 | – | TTL |
|---|---|---|---|---|---|---|---|
| | **Financial Indicators** | **Company Characteristics** | **Market-. Related Factors** | **Industry Classification** | **Shareholding Ratios** | **Time point Information** | |
| Reference | β 1 | β 2 | β 3 | β 4 | β 5 | β 6 | |
| Explanatory Variables | 9 | 8 | 4 | 33 | 5 | 1 | 60 |
| Base line | – | – | △1 | △1 | △1 | – | △3 |
| Not statistically significant | weak2 | – | △1 | △5 | weak2 | – | △10 |
| **Sub-TTL** | **7** | **8** | **2** | **27** | **2** | **1** | **47** |
| Result: Upward | 4 | 4 | – | 10 | 1 | – | 19 |
| Result: Downward | 1 | 2 | 2 | 17 | 1 | – | 23 |
| Result: Other | 2 | 2 | – | – | – | 1 | 5 |
| **Unexpected Findings** | **4** | **4** | **–** | **15** | **1** | **–** | **24** |

variables, this confirms the trend that larger Companies are more proactive in disclosure, which is not far above the expected range.

2. **β1-6: Operating Profit**

With respect to Operating Profit, some prior studies showed an association with positive voluntary disclosures, while others showed negative results. Operating Profit was statistically significant, but the impact was limited. It is suggested that short-term earnings fluctuations may make it difficult for Companies to implement a consistent disclosure strategy.

3. **β1-7: Price-to-book ratio (PBR)**

PBR is an indicator to evaluate the market value of a company, since it is calculated by dividing the stock price per share by the net assets per share. In this study, too, PBR showed statistically significant results, but the impact was negligible. It is possible that Companies with higher market values are more trusted and do not feel the need for additional disclosure.

4. **β1-8: Price-to-Earnings Ratio (PER)**

Since PER is calculated by dividing the stock price per share by earnings per share, a high PER value is an indicator that reflects a company with high growth expectations. Normally, a company with a high PER would be highly valued by the market, and as a result, disclosure would be more important. However, our results revealed the opposite tendency: firms with high PERs were less likely to voluntarily disclose, possibly because their stock prices were already supported by the market, reducing the perceived need for additional disclosure.

**5.2.2. Hypothesis H2: The impact of corporate characteristics on voluntary disclosure.** With respect to Company characteristics, the results differed from the predictions in β2-2 (Years since Establishment), β2-5 (Foreign Market), β2-7 (ISO 45001), and β2-8 (ISO 14001). A discussion of each variable is presented below.

1. **β2-2:** Years **since Establishment**

The effect of Company age was limited and consistent with previous studies [24,25].

2. **β2-5: Foreign market**

In prior studies, it was predicted that Companies relying on foreign markets would be more willing to make voluntary disclosures in order to respond to different regulatory environments and investor expectations. However, in this study, Foreign Market resulted in less voluntary disclosure, a new finding that contradicts expectations. The Foreign Market may be taking a cautious disclosure strategy into consideration of disclosure standards and risks.

3. **β2-7: ISO45001**

ISO 45001 is a certification for occupational health and safety, and this study found that being certified under ISO 45001 has a positive impact on voluntary disclosure in Japan. Unlike previous studies, this result is a new finding: obtaining ISO 45001 certification may emphasize a company's commitment to social responsibility.

4. **β2-8: ISO14001**

ISO 14001 is a certification for environmental management, and this study confirms that being certified under ISO 14001 has a positive impact on voluntary disclosure in Japan. This is a new finding, as it differs from the results of previous studies, and it is thought that the intention is to actively promote environmental responsiveness and enhance social reputation through ISO 14001.

**5.2.3. Hypothesis H3: The impact of the listing market on voluntary disclosure.** Hypothesis H3 examines the impact of listing markets on voluntary disclosure. The experimental results were in line with the hypothesis, confirming that

Companies listed on the Standard (β3-2), Growth (β3-3), and Local (β3-4) markets are more reluctant to make voluntary disclosures than Companies listed on the Prime market. This result is consistent with previous studies and no new findings were found in particular.

### 5.2.4. Hypothesis H4: The impact of industry classification on voluntary disclosure.
Industry-specific differences were identified, with results differing from forecasts in certain industries. The following is a discussion of these findings.

1. Manufacturing(from β4-4-4 to β4-4-19)

In the manufacturing sector, Chemicals (β4-4-7), Pharmaceuticals (β4-4-8), Rubber Products (β4-4-10), Nonferrous Metals (β4-4-13), and Precision Instruments (β4-4-18) positively impacted voluntary disclosures as expected. Foods (β4-4-4), Textiles and Apparels (β4-4-5), Oil and Coal Products (β4-4-9), Glass and Ceramics Products (β4-4-11), Iron and Steel (β4-4-12), Metal Products (β4-4-14), Transportation Equipment (β4-4-17), and Other Products (β4-4-19) were reluctant. Particularly in the Food and Oil & Coal industries, it is possible that environmental issues and regulatory compliance may have inhibited disclosure.

1. Transport, Information and Communications(from β4-6-21 to β4-6-25)

Marine Transportation (β4-6-22) and Air Transportation (β4-6-23) were active, while Land Transportation (β4-6-21) and Warehousing and Harbor Transportation (β4 -6-24) were passive. In Land Transportation, labor issues and safety may have inhibited disclosure, while in Warehousing and Harbor Transportation, confidentiality of trading terms may have influenced disclosure.

1. Commercial(from β4-7-26 to β4-7-27)

Wholesale Trade (B 4-7-26) and Retail Trade (B 4-7-27) were, contrary to expectations, reluctant. Retail Trade in particular may have been reluctant to disclose in order to avoid the risk of information leakage in a highly competitive market.

1. Finance and Insurance(from β4-8-28 to β4-8-31)

Securities and Commodities Futures (B 4-8-29) was reluctant, while Banks (B 4-8-28), Insurance (B 4-8-30), and Other Financing Business (B 4-8-31) were active. Uncertainty and regulation of financial instruments transactions are thought to be restraining factors for disclosure.

5. Services(β4-10-33)

Services (β4-10-33) was, contrary to expectations, reluctant. Services may be cautious about disclosure due to the intangible nature of the industry and the particular nature of the competitive environment.

### 5.2.5. Hypothesis H5: The impact of shareholder composition on voluntary disclosure.
The results of the study of the impact of shareholder composition confirmed the tendency for voluntary disclosure to be more reluctant, especially for companies with a high percentage of "Other Corporation" ownership. The following is a discussion of the results.

1. β5-2: Ownership Percentage of Other Corporations

It is suggested that Companies with a high percentage of ownership by corporate shareholders may be more cautious about information disclosure and may intend to minimize information sharing for strategic reasons. It is possible that among corporate shareholders, information is shared only internally or among shareholders, and there is less need for broad market disclosure. Furthermore, corporate shareholders are likely to be reluctant to disclose information to avoid the risk of undermining their competitive advantage, and they may have less need to seek additional public information because of their ability to communicate directly with management.

One limitation of the study is that the use of a large number of Explanation variables limited the detailed analysis for each hypothesis. We also recognize as a limitation that we were unable to fully take into account the differences in the content and

format of the reports, which are determined arbitrarily by the Companies. Furthermore, the fact that the time variable was treated as a dummy variable is also an issue, as the characteristics of the panel data could not be fully evaluated.

## 6. Conclusion

In this study, we analyzed the voluntary disclosure behavior of Japanese listed companies from various perspectives and made several important findings. In particular, based on the hypothesis that financial indicators influence the degree of voluntary disclosure (referred to as Hypothesis 1 (H1) in this paper), we clearly showed differences in the effects of price-to-book ratio (PBR) and price-to-earnings ratio (PER) on disclosure behavior. Based on the hypothesis that corporate characteristics influence the degree of voluntary disclosure (Hypothesis 2 (H2)), we identified differences in the impact of whether the company is listed on a foreign market or has obtained ISO certification. Furthermore, based on the hypothesis that the listing market influences the degree of voluntary disclosure (Hypothesis 3 (H3)), the effect of differences in listing market on disclosure behavior was clarified. Finally, based on the hypothesis that shareholder composition influences the degree of voluntary disclosure (Hypothesis 5 (H5)), the effect of shareholdings in other corporations on voluntary disclosure behavior was demonstrated.

The logistic regression analysis clearly visualizes the varying impacts of explanatory variables on voluntary disclosure, as presented in Fig 14. Variables are categorized into three groups based on their effects: "Upward," "Downward," and "Other." "Upward" variables increase the likelihood of voluntary disclosure, while "Downward" variables decrease this likelihood. The category "Other" indicates variables without a significant effect. The horizontal line at probability 0.5 indicates the threshold at which voluntary disclosure becomes more probable.

This chart provides an intuitive overview of the factors that influence disclosure behavior and can serve as a guide to which factors a company should focus on its disclosure strategy.

**In brief summary, the study yielded the following findings.**

1. **Impact of Financial Indicators**

   - The larger the **size of the company** (total assets and sales), the more voluntary disclosure tends to be promoted.

   - **Profitability** (e.g., operating margin, price-to-earnings ratio) has a complex effect on voluntary disclosure. A trend toward restrained disclosure was identified, especially for Companies with high PERs (price-to-earnings ratios).

2. **Impact of Company Characteristics**

   - The number of **employees** and **ISO certifications** (e.g., ISO 14001 and ISO 45001) are factors that actively promote voluntary disclosure.

   - On the other hand, **listing on a foreign market** acts as a factor that restrains disclosure behavior.

3. **Impact of Listed Markets**

   - Companies listed on the prime market are more active in voluntary disclosure due to the regulatory environment and transparency requirements from investors.

   - Voluntary disclosure tends to be restrained for companies listed on the Standard, Growth, and Local markets.

4. **Impact of Industry Classification**

   - A trend toward active voluntary disclosure was observed in certain industries, such as manufacturing and information and telecommunications.

   - On the other hand, there is a marked tendency for disclosure to be suppressed in agriculture, forestry, fisheries, mining, and local public service-related industries.

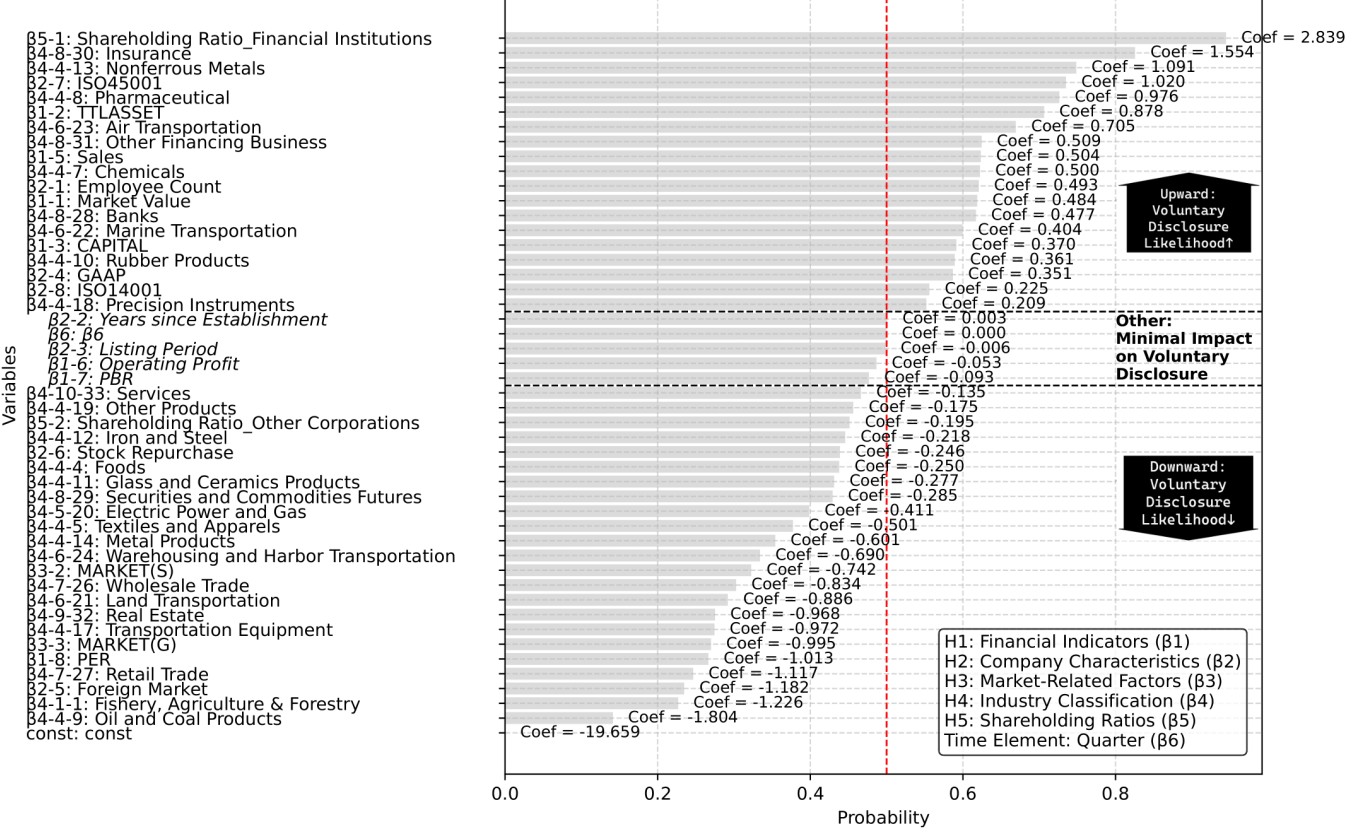

**Fig 14. Visualization of the probability & impact of voluntary disclosure using logistic regression.** This figure visually represents the probability and direction of influence each explanatory variable has on voluntary disclosure. Variables identified as "Upward" positively impact disclosure probability, making voluntary disclosure more likely as these variables increase. Conversely, "Downward" variables negatively impact disclosure probability, thus reducing disclosure likelihood as these variables rise. Variables categorized as "Other" do not exhibit a significant directional effect. The dotted line indicates a probability threshold (0.5), above which voluntary disclosure becomes increasingly likely.

5. **Impact of Shareholder Composition**

   • Voluntary disclosure is encouraged when a financial institution is a major shareholder.

   • For companies with many other corporate shareholders, disclosure is restrained to protect their competitive advantage.

6. **Other Findings**

   • **Regional characteristics**: companies listed only on local markets tend to be less active in voluntary disclosure.

   • **Chronological change**: The content of disclosure is changing as companies move toward integrated reporting and adopt new reporting formats.

   • **International comparison**: Compared to mandatory disclosure standards in other countries, Japanese companies have chosen to pursue an autonomous disclosure strategy.

As an academic contribution, this study adds a new perspective on the impact of voluntary disclosure based on industry classification and shareholder composition and reveals detailed industry differences that have not been captured in

previous studies. The study also delved into the previously unexplored impact of ISO certification and shareholder composition on corporate disclosure behavior. Furthermore, the method of classifying Explanation variables into "up," "down," and "other" shapes provides a new theoretical framework for voluntary disclosure behavior.

As a practical contribution, the study provides suggestions for companies to formulate disclosure strategies more effectively by understanding specifically which Explanation variables influence disclosure behavior. In particular, taking into account the degree of influence of each variable shown in Fig 7 will enable companies to formulate disclosure policies that are more in tune with the competitive environment and investor expectations. In addition, by clarifying the impact of ISO certification and differences in shareholder composition on disclosure behavior, it is possible to provide guidance on what strategies companies should adopt with respect to disclosure.

Additional research is needed to address the above new findings from future studies.

## Supporting information

**S1 File. Original dataset.** The original dataset was used in this study.
(CSV)

**S2 File. Explanation variables.** Detailed definitions and summary statistics for all explanatory variables used in the analysis.
(XLSX)

**S3 File. Overall evaluation of the logistic regression model.** Results of the logistic regression model evaluation, including accuracy, confusion matrix, classification report, and area under the ROC curve (AUC).
(DOCX)

**S4 File. Result of explanatory variables.** Detailed results of logistic regression analysis focusing on explanatory variables.
(XLSX)

**S5 File. Summary of experimental results with unexpected findings.** A summary of experimental results highlighting unexpected findings.
(XLSX)

## Author contributions

**Conceptualization:** Yuichiro Nakai.

**Data curation:** Yuichiro Nakai.

**Formal analysis:** Yuichiro Nakai.

**Investigation:** Yuichiro Nakai.

**Methodology:** Yuichiro Nakai, Mitsuo Yoshida.

**Project administration:** Yuichiro Nakai.

**Resources:** Yuichiro Nakai, Mitsuo Yoshida.

**Supervision:** Mitsuo Yoshida.

**Validation:** Yuichiro Nakai.

**Visualization:** Yuichiro Nakai.

**Writing – original draft:** Yuichiro Nakai.

**Writing – review & editing:** Mitsuo Yoshida.

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
