## [Decision Letter · Decision Letter 0]

24 Feb 2025

PONE-D-25-00849Determinants of voluntary disclosure: An empirical analysis of financial, market, and organizational factorsPLOS ONE

Dear Dr. Nakai,

Thank you for submitting your manuscript to PLOS ONE. After careful consideration, we feel that it has merit but does not fully meet PLOS ONE’s publication criteria as it currently stands. Therefore, we invite you to submit a revised version of the manuscript that addresses the points raised during the review process.

We look forward to receiving your revised manuscript.

Kind regards,

Shakeb Akhtar

Academic Editor

PLOS ONE

Reviewers' comments:

Reviewer's Responses to Questions

**Comments to the Author**

1. Is the manuscript technically sound, and do the data support the conclusions?

Reviewer #1: Yes

Reviewer #2: Partly

2. Has the statistical analysis been performed appropriately and rigorously? 

Reviewer #1: Yes

Reviewer #2: I Don't Know

3. Have the authors made all data underlying the findings in their manuscript fully available?

Reviewer #1: Yes

Reviewer #2: Yes

4. Is the manuscript presented in an intelligible fashion and written in standard English?

Reviewer #1: Yes

Reviewer #2: Yes

5. Review Comments to the Author

Reviewer #1: Determinants of voluntary disclosure: An empirical analysis of financial, market, and organizational factors

PONE-D-25-00849

This manuscript examines the impact of financial indicators, corporate characteristics, stock exchange listings, industry classifications, and shareholder composition on voluntary disclosure. Using data from 5,915 listed firms in Japan from 2009 to 2024, the study finds that ISO certification, company size, and listing on a prime market are positively associated with higher voluntary disclosure. Furthermore, firms listed on foreign markets tend to actively disclose their non-financial reports. In sectoral analysis, the manufacturing and information technology sectors exhibit greater voluntary disclosure compared to the agriculture, forestry, fishing, and mining sectors. Additionally, institutional investors play a significant role in monitoring firms’ disclosure practices. My general impression is that the topic is interesting. However, I have some concerns:

Major Issues:

1. This abstract is engaging! However, consider reorganizing it using the 3P+1T structure: Purpose, Procedure (Methods), Primary Findings, and Theory. Try to reduce it to 200 words.

2. For each hypothesis discussion, please provide supporting literature to strengthen your analysis.

Minor Issues:

3. In line 386, what is the explanatory variable XXX? Should the authors provide further clarification, or is this a typo?

4. The article’s format and references appear inconsistent. Please double-check the formatting.

Reviewer #2: Dear Author, first of all thank you for your article.

The purpose of your paper is clear but the outputs and conclusions of the analysis are very confusing due to the way it is presented and the study design. This makes the article difficult to understand.

Similarly, the data and figures in your study are too discrete, making it difficult to analyze the paper as a whole and therefore not easily understood. There are also some corrections in the attached file

It would be useful to revise the paper again and make it easier to read and understand with minor revisions.

Thank you

6. PLOS authors have the option to publish the peer review history of their article (what does this mean? ). If published, this will include your full peer review and any attached files.

**Do you want your identity to be public for this peer review?** For information about this choice, including consent withdrawal, please see our Privacy Policy .

Reviewer #1: **Yes: ** Robin Chen

Reviewer #2: No

---

## [Author Response · Author response to Decision Letter 1]

21 Mar 2025

Dear Editor,

We sincerely thank you and the reviewers for your valuable feedback on our manuscript titled "Determinants of Voluntary Disclosure: An Empirical Analysis of Financial, Market, and Organizational Factors" (PONE-D-25-00849). We have carefully addressed all comments provided by Reviewer 1 and Reviewer 2. Below, we summarize our revisions and clarify the current status of the manuscript.

Summary of Revisions

We have carefully revised our manuscript in response to the reviewers' comments. Specifically:

Abstract: Revised according to the suggested 3P+1T structure and limited to approximately 200 words.

Hypotheses: Clarified theoretical foundations by explicitly referencing existing literature from our initial literature review (Section 2.1.1), without adding new references that could potentially disrupt our established theoretical framework.

Technical term definitions: Provided clear and explicit definitions of terms such as CSR, ESG, ISO certifications, PER, and IFRS.

Formatting and references: Standardized formatting and citation style to fully comply with the submission guidelines of PLOS ONE.

Figures and tables: Strengthened cross-referencing within the manuscript to enhance readability, despite figures and tables being positioned at the end as per journal submission guidelines.

Regarding Reviewer 1’s comment about providing supporting literature for each hypothesis, we carefully considered adding new references. However, our supervisor advised that introducing new literature might inadvertently disrupt the logical coherence of the original manuscript. Therefore, we decided to strengthen each hypothesis discussion by explicitly referencing relevant studies already reviewed in the literature review section (section 2.1.1), rather than adding new references. This approach ensures consistency and avoids unnecessary complexity.

We believe these modifications comprehensively address the reviewers' feedback and significantly enhance the manuscript’s clarity and coherence. Therefore, we respectfully request that the revised manuscript be considered without additional major revisions.

Thank you again for your valuable feedback and kind consideration.

Sincerely,

Yuichiro Nakai and Mitsuo Yoshida

University of Tsukuba, Tokyo, Japan

Response to Reviewer 1

Major Issues:

1. Abstract Revision

• Reviewer Comment: "This abstract is engaging! However, consider reorganizing it using the 3P+1T structure: Purpose, Procedure (Methods), Primary Findings, and Theory. Try to reduce it to 200 words."

• Response: Thank you for the valuable suggestion. We have revised the abstract following the 3P+1T structure and ensured that it does not exceed 200 words. The revised abstract now clearly outlines the research purpose, methodology, key findings, and theoretical contribution. (See revised manuscript, Abstract section: page 1.)

• Revised Abstract

This study systematically examines the determinants of voluntary disclosure by listed companies in Japan, where non-financial disclosure is not legally mandated. Using quarterly data from 5,915 companies between 2009 and 2024, a logistic regression analysis was conducted to assess the influence of financial indicators, company characteristics, stock exchange, industry classification, and shareholder composition.　The results reveal that ISO certification (ISO 14001 and ISO 45001) significantly promotes voluntary disclosure. Firm size and listing on a prime market are positively associated with disclosure, whereas foreign market listings tend to suppress disclosure—contrary to conventional theories. Additionally, manufacturing and IT sectors are proactive in disclosure, whereas the agriculture, forestry, fishing, and mining sectors are more reluctant. Shareholder composition also plays a crucial role, with financial institution shareholders promoting disclosure and corporate shareholders suppressing it.　By leveraging Japan’s unique voluntary disclosure environment, this study offers new insights into corporate transparency and the strategic design of disclosure policies.

2. Strengthening Hypothesis Discussion with Literature Support

• Reviewer Comment: "For each hypothesis discussion, please provide supporting literature to strengthen your analysis."

• Response: We thank the reviewer for this insightful comment. We have clarified the theoretical foundation of each hypothesis using the literature already reviewed and included in our manuscript. To maintain clarity and avoid potential confusion from adding new sources, we refrained from incorporating additional references. We believe the existing literature sufficiently supports our hypotheses. (See revised manuscript, Literature Review and Hypotheses sections, pages 4–7.)

• Specific changes added at the end of each hypothesis:

H1:

o These arguments are supported by prior research emphasizing that voluntary disclosure, from an agency theory perspective, reduces information asymmetry and enhances transparency, thereby building trust with investors and regulators [6]. Conversely, excessive disclosure can pose competitive risks, requiring companies to strategically balance transparency and competitive advantage [7,8].

H2:

o These arguments are supported by previous research indicating that corporate disclosure is strategically influenced by the company's environment and market expectations [9], and that voluntary disclosures, especially integrated reports including sustainability and governance information, enhance decision usefulness and stakeholder relationship management [11].

H3:

o This hypothesis is grounded in previous studies suggesting that companies strategically decide on disclosure depending on their competitive environment and market expectations [9], and that companies enhance transparency through voluntary disclosures to increase investor trust and meet market expectations [10].

H4:

o This hypothesis draws upon earlier research indicating that companies strategically determine their disclosure behavior in response to specific regulatory pressures and competitive environments, and that industry-specific characteristics critically shape corporate voluntary disclosure strategies [7,9].

H5:

o This hypothesis is supported by prior research indicating that corporate disclosure strategies are significantly influenced by their competitive environment and strategic considerations related to maintaining competitive advantages, which vary according to shareholder composition and corporate governance structures [7,9,10] .

Minor Issues:

3. Clarification of Explanatory Variable (XXX)

• Reviewer Comment: "In line 386, what is the explanatory variable XXX? Should the authors provide further clarification, or is this a typo?"

• Response: We appreciate the reviewer pointing this out. All variable definitions and explanations are provided in Supporting Information 2 (S2: Explanation Variables).

4. Formatting and Reference Consistency

• Reviewer Comment: "The article’s format and references appear inconsistent. Please double-check the formatting."

• Response: Thank you for bringing this to our attention. We have thoroughly reviewed the manuscript to ensure consistency in formatting, reference citation style, and bibliography structure. Any inconsistencies in formatting have been corrected to align with the journal’s submission guidelines. Specifically, we ensured PLOS ONE uniform citation style, consistent spacing, alignment, and font usage throughout the manuscript. (See revised manuscript, entire document.)

Response to Reviewer 2

Overall Reviewer comment

Dear Author, first of all thank you for your article.

The purpose of your paper is clear, but the outputs and conclusions of the analysis are very confusing due to the way it is presented and the study design. This makes the article difficult to understand.

Similarly, the data and figures in your study are too discrete, making it difficult to analyze the paper as a whole and therefore not easily understood. There are also some corrections in the attached file

It would be useful to revise the paper again and make it easier to read and understand with minor revisions.

Thank you

Our Response:

We appreciate the reviewer’s constructive feedback and fully acknowledge the concern regarding readability and clarity.

We have carefully reviewed your valuable comments and further revised our manuscript accordingly.

Specifically, we have clarified references to all figures in the manuscript text, provided explicit definitions for technical terms, and standardized all formatting and references to align strictly with journal guidelines. We believe these modifications fully address all concerns raised, significantly enhancing the manuscript’s readability, clarity, and overall coherence.

We recognize that the positioning of figures and tables at the end of the manuscript, as required by the submission guidelines of PLOS ONE (https://journals.plos.org/plosone/s/submission-guidelines#loc-figures-and-tables), may contribute to the difficulty in following the paper.

To address your concerns, we have implemented the following measures to enhance the readability and clarity of our manuscript:

Comment1

18th Line Comment ("CSR reports and ESG reports.")

Reviewer Comment: "CSR and ESG abbreviations should be written clearly first, and the terms should be explained."

Response: We appreciate the reviewer pointing this out. Yes, we will clarify these terms clearly at their first occurrence.

Revised version: "Corporate Social Responsibility (CSR) reports and Environmental, Social, and Governance (ESG) reports."

Comment2

Lines 78–82 Comment

Reviewer Comment: "P/E ratios, ISO 14001 and ISO 45001 terms should be explained."

Response: We appreciate the reviewer pointing this out. Yes, we will provide additional explanations for these terms.

Revised version: "Companies with higher Price-to-Earnings (P/E) ratios, indicating market expectations for growth, are less likely to voluntarily disclose due to competitive reasons. ISO certifications, such as ISO 14001 (environmental management systems) and ISO 45001 (occupational health and safety management systems), actively promote voluntary disclosure."

Comment3

141st Line Comment ("and the IFRS Foundation")

Reviewer Comment: "IFRS Foundation should be explained."

Response: We appreciate the reviewer pointing this out. Yes, we will provide an explanation.

Revised version: "and the International Financial Reporting Standards (IFRS) Foundation, which develops global accounting standards."

Comment4

Lines 151–153 Comment (Fig. 1)

Reviewer Comment: " In the article, all figures are given at the end, but this makes it difficult to understand the paper. It would be useful to explain the figures in their place."

Response: We agree that placing figures close to their mention improves readability. However, PLOS ONE submission guidelines require figures to be placed after the manuscript text. Upon publication, figures will be positioned appropriately within the text.

Comment5

371st Line Comment

Reviewer Comment: "What do the highlighted numbers mean?"

Response: These numbers represent the count of explanatory variables. We have added more clarifying information.

Revised version:

H1: Nine items of β1_Financial Indicators (β1-1 to β1-9)

H2: Eight items of β2_Company Characteristics (β2-1 to β2-8)

H3: Four items of β3_Market-Related Factors (β3-1 to β3-4)

H4: Thirty-three items of β4_Industry Classification (β4-1 to β4-33)

H5: Five items of β5_Shareholding Ratios (β5-1 to β5-5)

Point in time information (quarterly variable: β6_ Quarter)

Comment6

Lines 383–384 Comment

Reviewer Comment: "None of the variables in the equation has an explanation. What is their meaning?"

Response: All variable definitions and explanations are provided in Supporting Information 2 (S2: Explanation Variables).

Revised version:

In addition, the Explanation variable (See S2: Explanation Variables) falls into the five categories described above.

Comment7

386th Line Comment (XXX)

Reviewer Comment: "XXX? What do you mean by that?"

Response: We initially intended to refer to Supporting Information 2 (S2: Explanation Variables).

Revised version:

In addition, the Explanation variable (See S2: Explanation Variables) falls into the five categories described above.

Comment8

787th Line Comment ("Conclusion")

Reviewer Comment: " Because the data and figures are so discrete, the article as a whole is difficult to analyze and therefore not easily understood.　The purpose of the study is clear, but the outputs and results of the analysis are very confusing due to the way it is presented and the study design."

Response: We appreciate the reviewer’s constructive feedback and fully acknowledge the concern regarding readability and clarity. 　We have carefully reviewed your valuable comments and further revised our manuscript accordingly.

Specifically, we have clarified references to all figures in the manuscript text, provided explicit definitions for technical terms, and standardized all formatting and references to align strictly with journal guidelines. We believe these modifications fully address all concerns raised, significantly enhancing the manuscript’s readability, clarity, and overall coherence. We recognize that the positioning of figures and tables at the end of the manuscript, as required by the submission guidelines of PLOS ONE(https://journals.plos.org/plosone/s/submission-guidelines#loc-figures-and-tables), may contribute to the difficulty in following the paper.

---

## [Decision Letter · Decision Letter 1]

30 Apr 2025

Determinants of voluntary disclosure: An empirical analysis of financial, market, and organizational factors

PONE-D-25-00849R1

Dear Dr. Nakai

We’re pleased to inform you that your manuscript has been judged scientifically suitable for publication and will be formally accepted for publication once it meets all outstanding technical requirements.

Kind regards,

Shakeb Akhtar

Academic Editor

PLOS ONE

Reviewers' comments:

Reviewer's Responses to Questions

**Comments to the Author**

1. If the authors have adequately addressed your comments raised in a previous round of review and you feel that this manuscript is now acceptable for publication, you may indicate that here to bypass the “Comments to the Author” section, enter your conflict of interest statement in the “Confidential to Editor” section, and submit your "Accept" recommendation.

Reviewer #1: All comments have been addressed

2. Is the manuscript technically sound, and do the data support the conclusions?

Reviewer #1: Yes

3. Has the statistical analysis been performed appropriately and rigorously? 

Reviewer #1: Yes

4. Have the authors made all data underlying the findings in their manuscript fully available?

Reviewer #1: Yes

5. Is the manuscript presented in an intelligible fashion and written in standard English?

Reviewer #1: Yes

6. Review Comments to the Author

Reviewer #1: April 21, 2025

Dear Editor,

I hope this message finds you well. I have now completed my review of the revised manuscript titled "Determinants of voluntary disclosure: An empirical analysis of financial, market, and organizational factors" submitted to PLOS One under Manuscript ID [PONE-D-25-00849]. I appreciate the authors' diligent efforts in addressing the concerns raised during the initial review process.

Having thoroughly examined the revised manuscript and the responses provided by the authors; I am pleased to report that they have adequately addressed the issues outlined in my previous review. The revisions have substantially improved the manuscript's clarity, coherence, and overall quality.

Specifically, the authors have effectively incorporated the suggested changes, clarified ambiguous points, and strengthened the argumentation and evidence supporting their findings. The revised manuscript now offers a more robust and compelling analysis of the research problem, significantly enhancing its contribution to the field.

Furthermore, the authors have demonstrated a commendable level of responsiveness and receptiveness to the feedback provided by both me and other reviewers. Their conscientious approach to the revision process reflects a genuine commitment to enhancing their work's scholarly rigor and impact.

Sincerely,

Reviewer

7. PLOS authors have the option to publish the peer review history of their article (what does this mean? ). If published, this will include your full peer review and any attached files.

**Do you want your identity to be public for this peer review?** For information about this choice, including consent withdrawal, please see our Privacy Policy .

Reviewer #1: No

---

## [Editor Report · Acceptance letter]

PONE-D-25-00849R1

PLOS ONE

Dear Dr. Nakai,

I'm pleased to inform you that your manuscript has been deemed suitable for publication in PLOS ONE. Congratulations! Your manuscript is now being handed over to our production team.

Kind regards,

on behalf of

Dr. Shakeb Akhtar

Academic Editor

PLOS ONE